# Co-complexity: An Extended Perspective on Generalization Error

## Abstract

It is well known that the complexity of a classifier's function space controls its generalization gap, with two important examples being VC-dimension and Rademacher complexity (R-Complexity). We note that these traditional generalization error bounds consider the ground truth label generating function (LGF) to be fixed. However, if we consider a scenario where the LGF has no constraints at all, then the true generalization error can be large, irrespective of training performance, as the values of the LGF on unseen data points can be largely independent of the values on the training data. To account for this, in this work, we consider an extended characterization of the problem, where the ground truth labels are generated by a function within another function space, which we call the *generator* space. We find that the generalization gap in this scenario depends on the R-Complexity of both the classifier and the generator function spaces. Thus, we find that, even if the R-Complexity of the classifier is low and it has a good training fit, a highly complex generator space could worsen generalization performance, in accordance with the no free lunch theorem. Furthermore, the characterization of a generator space allows us to model constraints, such as invariances (translation and scale in vision) or local smoothness. Subsequently, we propose a joint entropy-like measure of complexity between function spaces (classifier and generator), called co-complexity, which leads to tighter bounds on the generalization error in this setting. Co-complexity captures the similarities between the classifier and generator spaces. It can be decomposed into an invariance co-complexity term, which measures the extent to which the classifier respects the invariant transformations in the generator, and a dissociation co-complexity term, which measures the ability of the classifier to differentiate separate categories in the generator. Our major finding is that reducing the invariance co-complexity of a classifier, while maintaining its dissociation co-complexity, improves the training error and reduces the generalization gap. Furthermore, our results, when specialized to the previous setting where the LGF is fixed, lead to potentially tighter generalization error bounds. Theoretical results are supported by empirical validation on the CNN architecture and its transformation-equivariant extensions. Co-complexity showcases a new side to the generalization abilities of classifiers and can potentially be used to improve their design.

## 1 Introduction

In the context of supervised classification, a major factor for consideration is the generalization error of the classifier, i.e., how good a classifier generalizes to test (unseen) data points. The notion of overfitting describes the case when the test error significantly exceeds the training error. Naturally, the objective for building a robust classifier entails the minimization of this *generalization gap*, to avoid overfitting. To that end, statistical studies on generalization error (Blumer et al. (1989); Bartlett & Mendelson (2003)) find that *complexity* measures on the classifier function space, $\mathcal{F}$, often directly control the generalization gap of a classifier. Two prominent examples of such measures include the Rademacher Complexity $\mathcal{R}_m(\mathcal{F})$ (Bartlett & Mendelson (2003)) and the VC dimension (Blumer et al. (1989)) $VC(\mathcal{F})$. Both measures directly estimate the *flexibility* of a function space, i.e., how likely is it for $\mathcal{F}$ to contain functions that can fit any random labelling over a set of data points. In this paper, we work with Rademacher complexity and propose extensions that provide a new perspective on generalization error.

From a statistical perspective, the generalization gap can be understood through convergence bounds on the error function, i.e., the expected deviation of the error function on the test data compared to the

training data. Traditional generalization error bounds (Bartlett & Mendelson, 2003) state that function complexity (i.e., $\mathcal{R}_m(\mathcal{F})$) directly corresponds to the generalization gap. Thus, higher $\mathcal{R}_m(\mathcal{F})$ usually leads to a greater generalization gap and slower convergence. Although the original $\mathcal{R}_m(\mathcal{F})$ was proposed for binary classification, similar results have been shown for multi-class settings and a larger variety of loss functions (Xu et al., 2016; Liao et al., 2018). Note that $\mathcal{R}_m(\mathcal{F})$ is over the entire function space and thus *global* in nature. *Local* forms of Rademacher complexity, which involve restricting the function space and lead to minimum error on the training data samples, have been proposed (Bartlett et al., 2005; 2002). Apart from function complexity based measures, there is also considerable work which uses an information theoretic perspective (Xu & Raginsky, 2017; Russo & Zou, 2020; Bu et al., 2020; Haghifam et al., 2020) in treating the subject.

## 1.1 WHY THE GROUND TRUTH LABEL GENERATING FUNCTION MATTERS

We define the label generating function for a classification problem as the function which generates the true labels for all possible datapoints. Note that most generalization error bounds, including the traditional ones, primarily are introspective in nature, i.e., they consider the size and flexibility of the classifier's function space $\mathcal{F}$. The main direction proposed in this work is the investigation of the *unknowability* of the ground truth label generating functions (LGF), using another function space which we call the *generator* space.

The generalization error bounds in Bartlett & Mendelson (2003) state that the difference in test and training performance is roughly bounded above by the Rademacher Complexity of the classifier's function space (i.e., $\mathcal{R}_m(\mathcal{F})$). In other words, whatever the training error, the test error will always be likely to be greater by an amount $\mathcal{R}_m(\mathcal{F})$ on average. We note that, in deriving the original bound, a major assumption is that the LGF $g$ is fixed and knowable from the data.

We now outline our main argument for taking the generator space into account. The LGF is indeed fixed, i.e., there cannot be two different ground truth label generating functions applicable to the same problem. However, our primary emphasis is on the fact that the true LGF will always be *unknown*, i.e., for any finite training data containing data-label pairs $(z_i, g(z_i))$, we would only truly know the output of the label generating function on the given training data samples. Only when we have infinite training data samples, the values of LGF are known at each $z \in \mathcal{R}^d$. In this work, we denote the function space of all possible LGFs, within which the true LGF is contained, as the *generator* space. Note that the generator space arises due to the unknowability of the LGF. We show in this work that due to the generator space, the true generalization gap is greater than the Rademacher complexity of the classifier, and also depends on the Rademacher complexity of the generator. Note that the size of the generator space, which dictates its complexity, will be dependent on the amount of constraints that the LGF has: if the LGF has no constraints at all, then generator spaces are larger, whereas if the LGF is constrained to be smooth/invariant to many transformations, generator spaces are smaller (as the set of functions which are smooth and invariant are also much smaller, for instance in vision).

Let us consider the case where the LGF $g$ has no constraints at all, i.e., it is sampled from a generator space $G$ which contains all possible functions $g : \mathcal{R}^d \rightarrow \{-1, 1\}$. In this case, the function $g$ is expected to have no structure and behaves like a random function, and thus the expected test accuracy of *any* classifier will be 50% (i.e., random chance). Therefore, even if the classifier function $f \in \mathcal{F}$ produces a very good fit on the training data and $\mathcal{F}$ happens to have a low complexity measure $\mathcal{R}_m(\mathcal{F})$, the generalization performance will still be poor, as no knowledge of the LGF values on the unseen datapoints is available from the training samples. This is in contrast to the generalization error bounds based on Rademacher complexity, which would estimate that the classifier should have good generalization performance (i.e., low test error), as both $\mathcal{R}_m(\mathcal{F})$ and training error are low. Note that although a typical training dataset extracted from this LGF may be hard to fit using a low-complexity classifier $\mathcal{F}$, there will be examples of training instances with non-zero probability on which a low-complexity classifier can have a good fit. The takeaway from this example is that the structure of the data (here represented using the complexity of the generator space) can additionally dictate whether a classifier can generalize. Note that, in this scenario, the expected generalization performance would be better if the LGF had more structure.

Figure 1 illustrates the role of both generator and classifier spaces in generalization via the two example scenarios discussed earlier. In both examples, the same low-complexity classifier has a good fit on the training data, thus $\mathcal{R}_m(\mathcal{F})$ is low. In example (a), the LGF has no constraints; while, in example (b), the LGF has constraints such as local smoothness. In example (a), the classifier

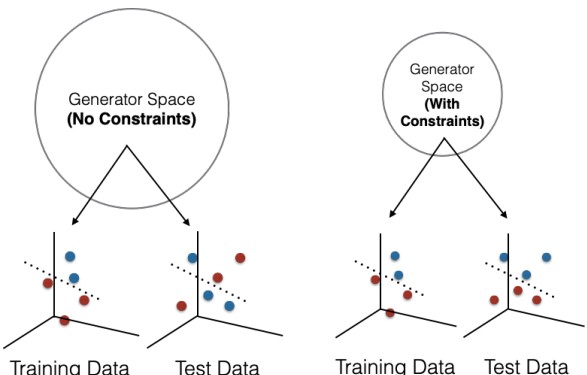

Figure 1: Two examples where the same low-complexity classifier shows similar fit on training samples, but in (a) the LGF has no constraints (large generator space) and in (b) the LGF is constrained by smoothness and other invariances (small generator space).

clearly exhibits poor generalization performance on the test data. This agrees with our previous argument that if the LGF has no constraints, generalization is essentially impossible. In example (b), the classifier shows much more robust generalization performance on the test data, due to the LGF having significantly more structure.

Furthermore, it is intuitively clear that a classifier function space $\mathcal{F}$ which has a high overlap with the generator space $\mathcal{G}$ (and therefore its constraints) should yield good generalization performance. This shows, that in addition to the function spaces $\mathcal{F}$ and $\mathcal{G}$ individually affecting the generalization gap, the similarities between $\mathcal{F}$ and $\mathcal{G}$ are also important. Both of these perspectives play leading roles in our construction of generator-and-classifier aware complexity measures and the associated novel generalization error bounds.

### 1.2 RELATED WORK

To the best of our knowledge, the approach we are proposing to study generalization performance is novel and there is not much directly related work. Here, we describe some examples of works which discuss relevant concepts. The no free lunch theorem proposed in Wolpert & Macready (1997) indirectly sheds light on the behaviour of the LGFs. However, it does not incorporate ways to reduce variability in the LGFs by considering constraints related to the classification problem. Invariance constraints in learning algorithms were studied in Sokolic et al. (2017), where the input space was factored into invariant transformations, similar to what we also do in this work. In doing so, the *complexity* of the data was indirectly explored, based on the number of invariant transformations present in the input space. However, the generalization bounds were derived with an assumption of perfectly invariant classifier function spaces, which is not applicable for CNNs and their variants (as shown in Kauderer-Abrams (2017)). In another relevant study (Jin et al. (2019)), a *cover-complexity* measure of a single dataset was proposed, and the generalization error of fully connected networks was analyzed with respect to the same. However, invariances in the dataset and the learning algorithm were neglected, i.e., the similarities between the generator and the classifier spaces were not studied.

### 1.3 KEY CONTRIBUTIONS

The contributions of this work are as follows:

1. We propose a novel complexity measure between the classifier function space $\mathcal{F}$ and the generator function space $\mathcal{G}$ called *co-complexity*, which we use to derive new, more accurate global estimates of *generator-aware* generalization error bounds (Theorem 3 and 4). Co-complexity considers not only the complexity of the generator and classifier function spaces, but also the similarities between them. Doing so allows for the a more exhaustive look into generalization error.

2. We decompose co-complexity into two different measures of complexity, invariance co-complexity and dissociation co-complexity, which are used to derive new generalization error bounds, including bounds on the expected training error (Theorem 6). We find that reducing invariance co-complexity while keeping the dissociation co-complexity unchanged, helps reduce the generalization gap (low variance) while maintaining low training error (low bias). This emphasizes

the importance of having classifiers that share invariance properties with generator spaces, e.g., rotation-invariant CNNs (Cohen & Welling (2016b)) on MNIST-Rot (Larochelle et al. (2007)).

3. We present empirical validation of co-complexity measures of CNN and its scale-equivariant and rotation-equivariant counterparts (SE-CNN in Sosnovik et al. (2019), RE-CNN in Cohen & Welling (2016b)), which explains their superior generalization ability compared to MLPs.

Our proposed error bounds are easily specialized to the case where ground truth label function is fixed, leading to potentially tighter generalization error bounds (see Appendix A). Although our proposed measures are *global* in nature, *local* variants can be derived via similar extensions used in Bartlett et al. (2005).

## 2 DEFINITIONS

Assume that we have $m$ number of $d$-dimensional i.i.d sampled instances $S = [z_1, z_2, .., z_m]$ drawn from some distribution $P$, and another set of $m$ i.i.d sampled instances $S' = [z'_1, z'_2, .., z'_m]$ also drawn from $P$. Define $\sigma = [\sigma_1, \sigma_2, .., \sigma_m]$ where $\sigma_i$ are i.i.d random variables following the Rademacher distribution ($Pr(\sigma_i = +1) = Pr(\sigma_i = -1) = 0.5$). $\mathcal{F}$ and $\mathcal{G}$ are two function spaces from $\mathbb{R}^d \to \{-1, 1\}$ and we assume that they are defined at all points in $\mathbb{R}^d$. In the context of our problem, $\mathcal{F}$ will be the classifier's function space, whereas $\mathcal{G}$ will be the generator space.

**Rademacher Complexity (Bartlett & Mendelson (2003)):** First, we provide the definition of *Rademacher Complexity*:

$$\mathcal{R}_m(\mathcal{F}) = \mathop{\mathbb{E}}_{\sigma, S} \left[ \sup_{f \in \mathcal{F}} \left( \frac{1}{m} \sum_{i=1}^{m} f(z_i) \sigma_i \right) \right] \tag{1}$$

It can be seen that $\mathcal{R}_m(\mathcal{F})$ indicates the noisy-label fitting ability of the classifier's function space $\mathcal{F}$.

**Correlated Rademacher Complexity:** We propose a modified form of the original Rademacher complexity, called *Correlated Rademacher Complexity*, $\mathcal{R}_m^C(\mathcal{F})$, which is defined as follows:

$$\mathcal{R}_m^C(\mathcal{F}) = \frac{1}{2} \times \mathop{\mathbb{E}}_{\sigma, S, S'} \left[ \sup_{f \in \mathcal{F}} \left( \frac{1}{m} \sum_{i=1}^{m} f(z_i) f(z'_i) \sigma_i \right) \right] \tag{2}$$

We have $0 \le \mathcal{R}_m^C(\mathcal{F}) \le 1/2$. We also show that $\mathcal{R}_m^C(\mathcal{F}) \le \mathcal{R}_m(\mathcal{F})$ (see Appendix C.1). It has been argued that $\mathcal{R}_m(\mathcal{F})$ (and therefore $\mathcal{R}_m^C(\mathcal{F})$) can be considered as "entropy" measures of the entire function space $\mathcal{F}$ (Anguita et al. (2014)). Note that, like $\mathcal{R}_m(\mathcal{F})$, $\mathcal{R}_m^C(\mathcal{F})$ is also eventually depends on the noisy label fitting ability of $\mathcal{F}$, but computes it via ability of the function space to assign the same or a different label to two random points $z_i$ and $z'_i$ (via $f(z_i) f(z'_i)$). Thus we denote it as the *correlated* Rademacher complexity of $\mathcal{F}$.

**Co-Complexity:** Now we propose various complexity measures between two separate function spaces $\mathcal{F}$ and $\mathcal{G}$. Similar to $\mathcal{R}_m(\mathcal{F})$ and $\mathcal{R}_m^C(\mathcal{F})$, these measures assess the noisy label fitting abilities of the union of the function spaces $\mathcal{F}$ and $\mathcal{G}$. In doing so, these measures compute a joint-entropy like metric over the two function spaces. First, we define the *Co-complexity* between $\mathcal{F}$ and $\mathcal{G}$ as follows.

$$\mathcal{R}_m(\mathcal{F}, \mathcal{G}) = \frac{1}{2} \times \mathop{\mathbb{E}}_{\sigma, S, S'} \left[ \sup_{f \in \mathcal{F}, g \in \mathcal{G}} \left( \frac{1}{m} \sum_{i=1}^{m} f(z_i) f(z'_i) g(z_i) g(z'_i) \sigma_i \right) \right] \tag{3}$$

Some of the properties of co-complexity are as follows:

**P1** $\mathcal{R}_m(\mathcal{F}, \mathcal{G}) = \mathcal{R}_m(\mathcal{G}, \mathcal{F})$, i.e., co-complexity is symmetric.

**P2** $\mathcal{R}_m(\mathcal{F}, \mathcal{G}) \ge \mathcal{R}_m^C(\mathcal{F})$ and $\mathcal{R}_m(\mathcal{F}, \mathcal{G}) \ge \mathcal{R}_m^C(\mathcal{G})$, i.e., the co-complexity between $\mathcal{F}$ and $\mathcal{G}$ is always greater than the individual correlated Rademacher complexities of $\mathcal{F}$ and $\mathcal{G}$.

**P3** $\mathcal{R}_m(\mathcal{F}, \mathcal{G}) \le \mathcal{R}_m^C(\mathcal{F}) + \mathcal{R}_m^C(\mathcal{G}) \le \mathcal{R}_m(\mathcal{F}) + \mathcal{R}_m(\mathcal{G})$, i.e., the co-complexity between $\mathcal{F}$ and $\mathcal{G}$ is upper bounded by the sum of the Rademacher complexities of $\mathcal{F}$ and $\mathcal{G}$.

**P4** $\mathcal{R}_m(\mathcal{F}, \mathcal{G})$ behaves like a joint-entropy measure of $\mathcal{F}$ and $\mathcal{G}$. To see this, let us define $\mathcal{I}_m(\mathcal{F}, \mathcal{G}) = \mathcal{R}_m^C(\mathcal{F}) + \mathcal{R}_m^C(\mathcal{G}) - \mathcal{R}_m(\mathcal{F}, \mathcal{G})$, called the *mutual co-complexity* between $\mathcal{F}$ and $\mathcal{G}$. We later find that $\mathcal{I}_m(\mathcal{F}, \mathcal{G})$ behaves like a mutual-information measure. This, coupled with the fact that $\mathcal{R}_m^C(\mathcal{F})$ and $\mathcal{R}_m^C(\mathcal{G})$ can be construed as entropy measures of $\mathcal{F}$ and $\mathcal{G}$, implies the result.

**Invariance Co-Complexity:** Next, we quantify some of the properties of the ground truth generator space $\mathcal{G}$ in terms of its invariance transformations. We define the *Invariance Classes* of $\mathcal{G}$ as follows:

$$\mathcal{I}_\mathcal{C}(\mathcal{G}) = \{\tau_1(.,\theta_1), \tau_2(.,\theta_2), ..., \tau_n(.,\theta_n)\}, \tag{4}$$

where $\{\tau_i(.,\theta_i)|i = 1, 2, \cdots, n\}$ are functions from $\mathbb{R}^d \to \mathbb{R}^d$, and $\theta_i$ represents the extent of the transformation $\tau_i$. Each $\theta_i \in \mathbb{R}$ is a scalar and takes on a set of admissible values depending on the transformation (possibly infinite). The functions are constrained such that, $\forall \tau \in \mathcal{I}_\mathcal{C}(\mathcal{G}), \forall g \in \mathcal{G}$, $g(\tau(z), t) = g(z)$ for all data points $z \in \mathbb{R}^d$, and for all admissible values of the transformation parameter $t$. Also, note that setting $\theta_i = 0$ leads to the identity function, i.e., $\tau_i(z, 0) = 1$ for all $z \in \mathcal{R}^d$ and all $i$.

Based on the above we define a transformation-indicator function $I(z, z_i)$, such that $I(z, z_i) = 1$ if $z = \tau_1(., t_1) \circ \tau_2(., t_2) \circ .... \circ \tau_n(., t_n)(z_i)$, for a certain $t_1, t_2, ...t_n$ and otherwise $I(z, z_i) = 0$. Additionally, with respect to a generator space $\mathcal{G}$, we also define the *invariance extended set* of a datapoint $z_i \in \mathbb{R}^d$ as follows:

$$\mathcal{I}_\mathcal{G}(z_i) = \{z \in \mathbb{R}^d \mid I(z, z_i) = 1\} \tag{5}$$

Note that that invariance extended set of $z$ will always have the same ground truth label as $z$. Next, we define the *Invariance Co-complexity* between two function spaces $\mathcal{F}$ and $\mathcal{G}$ as follows:

$$\mathcal{R}_m^I(\mathcal{F}, \mathcal{G}) = \frac{1}{2} \times \underset{S,S'}{\mathbb{E}} \left[ \sup_{f \in \mathcal{F}} \left( 1 - \left( \frac{1}{m} \sum_{i=1}^{m} f(z_i)f(z_i') \right) \right) \right], \text{where } z_i' \sim P(z)I(z, z_i), \forall i . \tag{6}$$

Note that $\mathcal{R}_m^I(\mathcal{F}, \mathcal{G}) \leq 1$. Each data point $z_i'$ in $S'$ is contained within the invariance extended set $\mathcal{I}_\mathcal{G}(z_i)$ of $z_i$, and is sampled according to the un-normalized distribution $P(z)I(z, z_i)$ over $z \in \mathbb{R}^d$. The invariance co-complexity between $\mathcal{F}$ and $\mathcal{G}$ indicates the degree to which $\mathcal{F}$ obeys the invariance transformations within $\mathcal{G}$. For instance, a low $\mathcal{R}_m^I(\mathcal{F}, \mathcal{G})$ would indicate that $f(z_i)f(z_i') = 1$ (i.e., $f(z_i) = f(z_i')$) for most $z_i$ and $z_i'$ which are related via some invariance transformation in $\mathcal{G}$.

**Dissociation Co-Complexity:** We now define the *Dissociation Co-complexity* between $\mathcal{F}$ and $\mathcal{G}$ when the corresponding datapoints in $S$ and $S'$ are not related by any invariance transformation. That is, for all $z_i \in S$ and $z_i' \in S'$, $z_i' \notin \mathcal{I}_\mathcal{G}(z_i)$, and is sampled from the distribution $P(z)(1 - I(z, z_i))$ over $z \in \mathbb{R}^d$. The dissociation co-complexity can then be defined as:

$$\mathcal{R}_m^D(\mathcal{F}, \mathcal{G}) = \frac{1}{2} \times \underset{\sigma,S,S'}{\mathbb{E}} \left[ \sup_{f \in \mathcal{F}} \left( \frac{1}{m} \sum_{i=1}^{m} f(z_i)f(z_i')\sigma_i \right) \right], \text{where } z_i' \sim P(z)(1 - I(z, z_i)), \forall i. \tag{7}$$

Note that $\mathcal{R}_m^D(\mathcal{F}, \mathcal{G})$ measures the average flexibility of $\mathcal{F}$ in its label assignment to any two points $z_i$ and $z_i'$ which are not related via any invariance transformation in $\mathcal{G}$. Thus, a larger $\mathcal{R}_m^D(\mathcal{F}, \mathcal{G})$ would indicate that the classifier function space $\mathcal{F}$ is able to easily assign separate labels to datapoints which are not related via any invariance transformation in $\mathcal{G}$. We also define a variant of $\mathcal{R}_m^D(\mathcal{F}, \mathcal{G})$, in which $S$ contains only one instance, instead of $m$ instances, denoted as $\mathcal{R}_m^{D,1}(\mathcal{F}, \mathcal{G})$. $\mathcal{R}_m^{D,1}(\mathcal{F}, \mathcal{G})$ is defined as follows.

$$\mathcal{R}_m^{D,1}(\mathcal{F}, \mathcal{G}) = \frac{1}{2} \times \underset{\sigma,z_0 \sim P,S'}{\mathbb{E}} \left[ \sup_{f \in \mathcal{F}} \left( \frac{1}{m} \sum_{i=1}^{m} f(z_0)f(z_i')\sigma_i \right) \right], \text{where } z_i' \sim P(z)(1 - I(z, z_0)), \forall i. \tag{8}$$

Note that $\mathcal{R}_m^D(\mathcal{F}, \mathcal{G}) \leq 0.5$ and $\mathcal{R}_m^{D,1}(\mathcal{F}, \mathcal{G}) \leq 0.5$.

**Rademacher Smoothness:** For our final result in Theorem 6, we assume that the function space $\mathcal{F}$ is Rademacher smooth w.r.t. $\mathcal{G}$. We define *Rademacher Smoothness* as follows. If $\mathcal{F}$ is Rademacher smooth w.r.t. $\mathcal{G}$, then any quantity of the form $\mathbb{E}_{S,S'}\left[\sup_{f \in \mathcal{F}}\left(\frac{1}{m}\sum_{i=1}^m f(z_i)f(z_i')g(z_i)g(z_i')\sigma_i\right)\right]$ or $\mathbb{E}_{S,S'}\left[\sup_{f \in \mathcal{F}, g \in \mathcal{G}}\left(\frac{1}{m}\sum_{i=1}^m f(z_i)f(z_i')g(z_i)g(z_i')\sigma_i\right)\right]$, where the expectation is over datapoint permutations $S, S'$, should lie between the two following cases : (i) $S' \in \mathcal{I}_\mathcal{G}(S)$ (i.e., $z_i' \sim P(z)(I(z, z_i))$ for all $i$) and (ii) $S' \notin \mathcal{I}_\mathcal{G}(S)$ (i.e., $z_i' \sim P(z)(1 - I(z, z_i))$ for all $i$). Then, for some non-zero value

of $\alpha$, $0 \leq \alpha \leq 1$, the following holds:

$$\underset{S,S'}{\mathbb{E}} \left[ \sup_{f \in \mathcal{F}} \frac{1}{m} \left( \sum_{i=1}^{m} f(z_i)f(z_i')g(z_i)g(z_i')\sigma_i \right) \right]$$

$$= (\alpha) \underset{S,S',S' \in \mathcal{I}_{\mathcal{G}}(S)}{\mathbb{E}} \left[ \sup_{f \in \mathcal{F}} \left( \frac{1}{m} \sum_{i=1}^{m} f(z_i)f(z_i')g(z_i)g(z_i')\sigma_i \right) \right]$$

$$+ (1 - \alpha) \underset{S,S',S' \notin \mathcal{I}_{\mathcal{G}}(S)}{\mathbb{E}} \left[ \sup_{f \in \mathcal{F}} \left( \frac{1}{m} \sum_{i=1}^{m} f(z_i)f(z_i')g(z_i)g(z_i')\sigma_i \right) \right]. \tag{9}$$

Similarly, the same applies to $\mathbb{E}_{S,S'} \left[ \sup_{f \in \mathcal{F}, g \in \mathcal{G}} \left( \frac{1}{m} \sum_{i=1}^{m} f(z_i)f(z_i')g(z_i)g(z_i')\sigma_i \right) \right]$.

## 3 THEORETICAL RESULTS

We now present a series of results which extend the generalization framework using generator spaces. The proofs of all our results are available in Appendix C. For the following results, we assume that the classifier's function space $\mathcal{F}$ contains the constant function $f(z) = c \; \forall z$, $c \in \{-1, 1\}$.

We use the definitions provided in the previous section. Specifically, $\mathcal{F}$ is be the classifier's function space and $\mathcal{G}$ is the generator space. Given that the data labels are generated by $g \in \mathcal{G}$, we extend the definition of the sampled instances $S$ by adding the output labels, i.e., $S = [(z_1, g(z_1)), (z_2, g(z_2)), .., (z_m, g(z_m))]$. Then for $f \in \mathcal{F}$, we denote the 0-1 loss on $S$ by

$$\widehat{err}_S(f) = \sum_{i=1}^{m} \frac{(1 - f(z_i)g(z_i))}{2m}. \tag{10}$$

We also denote the generalization error over the data samples generated by distribution $P$ as

$$err_P(f) = \lim_{N \to \infty, z_i \sim P} \sum_{i=1}^{N} \frac{(1 - f(z_i)g(z_i))}{2N}. \tag{11}$$

First, we describe the generalization error bound originally proposed in Bartlett & Mendelson (2003).

**Theorem 1.** (**Bartlett & Mendelson (2003)**) *For $0 < \delta < 1$, with probability $p \geq 1 - \delta$, we have*

$$err_P(f) \leq \widehat{err}_S(f) + \mathcal{R}_m(\mathcal{F}) + \sqrt{\frac{\log(1/\delta)}{2m}}. \tag{12}$$

The above theorem assumes that the ground truth function is completely knowable from the given training examples, and therefore the generator space contains only a single element (Bartlett & Mendelson (2003)). For all of our results that follow, we assume the extended case where the generator space can contain more than one element due to the unknowability of the LGF, and construct generalization error bounds that consider the complexity of $\mathcal{G}$ as well. Also note that the following results hold for every choice of $f \in \mathcal{F}$ and $g \in \mathcal{G}$. First we present the extended Theorem 1 when the LGF is unknowable ($\mathcal{R}_m(\mathcal{G}) > 0$).

**Theorem 2.** *For $0 < \delta < 1$, with probability $p \geq 1 - \delta$, we have,*

$$err_P(f) \leq \widehat{err}_S(f) + \mathcal{R}_m(\mathcal{F}) + \mathcal{R}_m(\mathcal{G}) + \sqrt{\frac{\log(1/\delta)}{2m}}. \tag{13}$$

**Remark 1.** *This result states that the complexity of the generator space also directly contributes towards increasing the generalization gap. In problems where the LGF is known to be heavily constrained and structured, $\mathcal{R}_m(\mathcal{G})$ will be low, reducing the expected generalization gap of classifiers, and vice-versa. Note that in the knowable ground truth scenario where $\mathcal{G}$ has a single element, we have $\mathcal{R}_m(\mathcal{G}) = 0$, which returns the error bound in Theorem 1. This result shows that larger complexity of $\mathcal{G}$ results in a greater generalization gap. We demonstrate this with neural network generator spaces of varying complexity (see Appendix B.3). Intuitively, the additional $\mathcal{R}_m(\mathcal{G})$ term comes from the fact that given the training data labels, there still exists a subspace of functions in $\mathcal{G}$ which are potential candidates for the ground truth function.*

The following result presents tightens Theorem 2 using co-complexity.

**Theorem 3.** *For $0 < \delta < 1$, with probability $p \geq 1 - \delta$, we have*

$$err_P(f) \leq \widehat{err}_S(f) + \mathcal{R}_m(\mathcal{F}, \mathcal{G}) + \sqrt{\frac{\log(1/\delta)}{2m}}. \tag{14}$$

**Remark 2.** *Co-complexity measures the degree of similarity between $\mathcal{F}$ and $\mathcal{G}$, instead of simply adding their respective complexities, and therefore is tighter than Theorem 2 (co-complexity **P3**). We also find that the generalization gap, for the case when the roles are reversed, is unchanged (see corollary 3.1 in Appendix C). This demonstrates an inherent symmetry in the problem. Also note that Theorem 3 leads to potentially tighter bounds when specialized to the conventional setting where $\mathcal{R}_m(\mathcal{G}) = 0$ (Appendix A).*

The following result outlines a lower bound on the generalization error, using co-complexity.

**Theorem 4.** *For $0 < \delta < 1$, with probability $p \geq 1 - \delta$,*

$$err_P(f) \geq \widehat{err}_S(f) - \mathcal{R}_m(\mathcal{F}, \mathcal{G}) - \sqrt{\frac{\log(1/\delta)}{2m}}. \tag{15}$$

**Remark 3.** *This result, coupled with Theorem 3, demonstrates that if one interprets the $\widehat{err}_S(f)$ term as the bias of the classifier, $\mathcal{R}_m(\mathcal{F}, \mathcal{G})$ can be interpreted as the variance.*

The following result addresses the joint-entropy like behaviour of $\mathcal{R}_m(\mathcal{F}, \mathcal{G})$.

**Theorem 5.** *We are given the mutual complexity measure $\mathcal{I}_m(\mathcal{F}, \mathcal{G})$ as defined in co-complexity **P4**. Consider another ground truth generator space $\mathcal{G}'$, such that $\mathcal{I}_m(\mathcal{G}', \mathcal{G}) = 0$, i.e., $\mathcal{G}$ and $\mathcal{G}'$ are independent. Then we have,*

$$\mathcal{R}_m(\mathcal{F}) \geq \mathcal{I}_m(\mathcal{F}, \mathcal{G}) + \mathcal{I}_m(\mathcal{F}, \mathcal{G}'). \tag{16}$$

**Remark 4.** *Let $H(X)$ represents the Shannon entropy of the random variable $X$ and $I(X; Y)$ represents the mutual information between random variables $X$ and $Y$. Then, it is known that $H(X) \geq I(X; Y) + I(X; Y')$, when $Y$ and $Y'$ are independent random variables ($I(Y; Y') = 0$). This observation, coupled with the fact that $\mathcal{R}_m^C(\mathcal{F})$ can be considered as an entropy measure of the function space (see section 2) indicates that the quantity $\mathcal{I}_m(\mathcal{F}, \mathcal{G})$ behaves like a mutual information estimate between the function spaces $\mathcal{F}$ and $\mathcal{G}$. Furthermore, as $\mathcal{R}_m(\mathcal{F}, \mathcal{G}) = \mathcal{R}_m^C(\mathcal{F}) + \mathcal{R}_m^C(\mathcal{G}) - \mathcal{I}_m(\mathcal{F}, \mathcal{G})$, we can see that $\mathcal{R}_m(\mathcal{F}, \mathcal{G})$ behaves as a joint entropy measure of two function spaces.*

The following result demonstrates how the co-complexity measure $\mathcal{R}_m(\mathcal{F}, \mathcal{G})$ can be decomposed into separate co-complexity measures.

**Lemma 1.** *Consider function spaces $\mathcal{F}$ and $\mathcal{G}$, such that $\mathcal{F}$ is Rademacher smooth w.r.t. $\mathcal{G}$. Let us define $\mathcal{R}_m^{C,D}(\mathcal{G})$ as $\frac{1}{2} \times \mathbb{E}_{\sigma, S, S', S' \notin \mathcal{I}_\mathcal{G}(S)} \left[ \sup_{g \in \mathcal{G}} \left( \frac{1}{m} \sum_{i=1}^m g(z_i) g(z_i') \sigma_i \right) \right]$. For some non-negative real constant $0 \leq \alpha \leq 1$, we then have*

$$\mathcal{R}_m(\mathcal{F}, \mathcal{G}) \leq \alpha \mathcal{R}_m^I(\mathcal{F}, \mathcal{G}) + (1 - \alpha)(\mathcal{R}_m^D(\mathcal{F}, \mathcal{G}) + \mathcal{R}_m^{C,D}(\mathcal{G})), \tag{17}$$

*where $\mathcal{R}_m^I(\mathcal{F}, \mathcal{G})$ and $\mathcal{R}_m^D(\mathcal{F}, \mathcal{G})$ are the invariance and dissociation co-complexity, respectively.*

**Remark 5.** *This decomposition allows us to differentiate the impact of $\mathcal{R}_m^I(\mathcal{F}, \mathcal{G})$ and $\mathcal{R}_m^D(\mathcal{F}, \mathcal{G})$ on the generalization error bound. Note that value of $\alpha$ here will be proportional to the cardinality of the invariance transformation classes $\mathcal{I}_\mathcal{G}$ of $\mathcal{G}$.*

Using this, we proceed to our final result, where we express the generalization error bound in Theorem 3, in terms of the invariance and dissociation co-complexities. For purposes of simplification, we denote $R_m^I(\mathcal{F}, \mathcal{G})$ as $R_m^I$, $R_m^D(\mathcal{F}, \mathcal{G})$ as $R_m^D$ and similarly for $\mathcal{R}_m^{D,1}$.

**Theorem 6.** *Consider function spaces $\mathcal{F}$ and $\mathcal{G}$, such that $\mathcal{F}$ is Rademacher smooth w.r.t. $\mathcal{G}$. Let $err_P$ denote the generalization error for the functions $f \in \mathcal{F}$ which showcase the best fit on the training data samples (averaged across all $g$ in $\mathcal{G}$). $\mathcal{R}_m^{C,D}(\mathcal{G})$ is defined in Lemma 1. For some non-negative real constants $0 \leq \alpha, \beta \leq 1$ and $0 < \delta < 1$, with probability $p \geq 1 - \delta$, we have*

$$err_P \leq (1 - \beta) \left( \frac{1}{2} - \mathcal{R}_m^{D,1} \right) + \alpha \mathcal{R}_m^I + (1 - \alpha)(\mathcal{R}_m^D + \mathcal{R}_m^{C,D}(\mathcal{G})) + \sqrt{\frac{2\log(1/\delta)}{m}} \tag{18}$$

**Remark 6.** *The first term in the generalization error bound in* (18) *represents an upper bound on the average training error of the function which best fits the training data. We find that smaller $R^I$ leads to a smaller generalization gap (variance), while keeping the training error unchanged. However, the same cannot be said for the dissociation co-complexity $R^D$. Thus, for a fixed $R^D$ and $R^{D,1}$, classifier function spaces with smaller $R^I$ will lead to smaller generalization error. Also note that when the generator space only contains a single element, $\mathcal{R}_m^{C,D}(\mathcal{G}) = 0$.*

The next proposition gives interpretable bounds on invariance and dissociation co-complexity. Let $\mathbf{z} = z_1, z_2, ..., z_m$, and $\mathbf{z'} = z'_1, ..., z'_m$, where $z_k, z'_k \in \mathbb{R}^d \; \forall \; k$. Define the well known growth function of binary valued functions $\mathcal{F}$ as $A = \Pi_{\mathcal{F}}(m) = \max_{\mathbf{z}} |\{f(z_k) : k = 1, 2, ..., m, f \in \mathcal{F}\}|$. Define the invariance-constrained and dissociation-constrained growth functions of $\mathcal{F}$ w.r.t. $\mathcal{G}$ as:

$$B = \Pi_{\mathcal{F}\,|\,\mathcal{G}}^I(m) = \max_{\mathbf{z}, \mathbf{z'}, z'_i \in \mathcal{I}_{\mathcal{G}}(z_i), \forall i} |\{f(z_k), f(z'_k) : k = 1, 2, ..., m, f \in \mathcal{F}\}|$$

$$C = \Pi_{\mathcal{F}\,|\,\mathcal{G}}^D(m) = \max_{\mathbf{z}, \mathbf{z'}, z'_i \notin \mathcal{I}_{\mathcal{G}}(z_i), \forall i} |\{f(z_k), f(z'_k) : k = 1, 2, ..., m, f \in \mathcal{F}\}|$$

**Proposition 1.** *Define* $\mathcal{R}_m^{I'}(\mathcal{F}, \mathcal{G}) = \frac{1}{2} \times \mathbb{E}_{\sigma, S, S', S' \in \mathcal{I}_{\mathcal{G}}(S)} \left[\sup_{f \in \mathcal{F}} \left(\frac{1}{m} \sum_{i=1}^m f(z_i) f(z'_i) \sigma_i\right)\right]$, *where* $z'_i \in \mathcal{I}_{\mathcal{G}}(z_i), \forall i$. *Note that* $\mathcal{R}_m^{I'}(\mathcal{F}, \mathcal{G}) \leq \mathcal{R}_m^I(\mathcal{F}, \mathcal{G})$. *Given the definitions above, we have* $\mathcal{R}_m^{I'}(\mathcal{F}, \mathcal{G}) \leq \sqrt{\frac{\log B}{2m}} \leq \sqrt{\frac{\log A}{m}}$ *and* $\mathcal{R}_m^D(\mathcal{F}, \mathcal{G}) \leq \sqrt{\frac{\log C}{2m}} \leq \sqrt{\frac{\log A}{m}}$.

**Remark 7.** *Reducing* $\Pi_{\mathcal{F}\,|\,\mathcal{G}}^I(m)$ *while keeping* $\Pi_{\mathcal{F}\,|\,\mathcal{G}}^D(m)$ *unchanged would result in low invariance co-complexity without affecting dissociation co-complexity, which can't be achieved by simply reducing* $\Pi_{\mathcal{F}}(m)$ *(or the* $VC(\mathcal{F})$*). This shows that although* $\mathcal{R}_m^I$ *and* $\mathcal{R}_m^D$ *are not completely independent, careful construction of* $\mathcal{F}$ *makes it possible to reduce* $\mathcal{R}_m^I$ *while maintaining* $\mathcal{R}_m^D$.

## 4 EXPERIMENTS AND DISCUSSIONS

Our experiments (on MNIST and STL-10) explore the implications of Theorem 6, which states that low $R_m^I$ while maintaining $R_m^D$ improves generalization while keeping training error unchanged. As all of our proposed complexity metrics assume binary classifiers, we create subsets of these datasets containing only two randomly chosen categories, which converts the problem into a binary classification scenario.

### 4.1 MEASURING INVARIANCE CO-COMPLEXITY

As invariance co-complexity is defined w.r.t. a transformation class ($\mathcal{I}_{\mathcal{G}}$), we compute the invariance co-complexity (in (6)) of various networks shown in Table 1 for four different transformations. For each transformation $\tau$, we choose the datapoint pairs $S$ and $S'$ in (6) such that $z'_i = \tau(z_i, t)$, where $t$ is the transformation parameter. The parameter $t$ is randomly chosen as follows: 0°-180° for rotation, 0.7-1.2 for scale, 0°-90° for shear angle and 0.5-4 pixels for translation. The computed $R_m^I$ values indicate the extent to which these networks naturally allow for invariance to various transformation types. To approximately compute $R_m^I$, we take 1000 randomly weighted networks to construct $\mathcal{F}$ to compute (6). We average the results over 100 batches of data ($S$ and $S'$), each containing 1000 examples ($m = 1000$) from their respective datasets. We compute $R_m^I$ for multi-layered perceptrons (MLPs), CNNs and their transformation-equivariant extensions: scale-equivariant CNN (SE-CNN in Sosnovik et al. (2019)) and rotation-equivariant CNN (RE-CNN in Cohen & Welling (2016b)). Note that the scale-equivariant CNN and the rotation-equivariant CNN are known to outperform the vanilla CNN on MNIST and its rotation and scale extensions (Sosnovik et al. (2019); Cohen & Welling (2016b)). Hence, these transformation-equivariant extensions showcase better generalization performance. One of the objectives of this experiment is to verify whether the invariance and dissociation co-complexities of these networks indeed point to better generalization performance, as is observed in literature. Please see Appendix D for architecture details.

In every case in Table 1, we find that CNNs have lower $R_m^I$ than MLPs. As expected, we see that SE-CNN and RE-CNN have even lower $R_m^I$. CNNs are primarily expected to be robust to translation, due to max-pooling layers. However, even for other transformations such as rotation and shear, the $R_m^I$ of the CNN is less than that of an MLP. Please see Appendix B.2 for a more detailed analysis on how invariance co-complexity varies with choice of the transformation parameter.

| Architecture | Invariance Co-Complexity ($\mathcal{R}_m^I$) | | | | | | | |
|---|---|---|---|---|---|---|---|---|
| | Rotation | | Scale | | Shear | | Translation | |
| | MNIST | STL-10 | MNIST | STL-10 | MNIST | STL-10 | MNIST | STL-10 |
| MLP | 0.424 | 0.393 | 0.2585 | 0.2545 | 0.409 | 0.386 | 0.425 | 0.2565 |
| CNN | 0.3875 | 0.375 | 0.2435 | 0.2205 | 0.3885 | 0.3725 | 0.365 | 0.213 |
| SE-CNN | n/a | n/a | 0.1845 | 0.1905 | 0.385 | 0.3535 | n/a | n/a |
| RE-CNN | 0.357 | 0.3455 | n/a | n/a | 0.3385 | 0.3545 | n/a | n/a |

Table 1: Mean invariance co-complexity values for all 4 networks over 4 transformations, rotation, scale, shear, and translation. We approximate the computation of the invariance co-complexity in (6) by taking 1000 randomly initialized networks in each case (for the supremum in (6)). For fairness, all architectures share approximately the same number of parameters (around 16k).

| | MLP | CNN | RE-CNN | SE-CNN |
|---|---|---|---|---|
| Dissociation Co-Complexity (MNIST) | 0.4415 | 0.44 | 0.438 | 0.438 |
| Dissociation Co-Complexity (STL-10) | 0.4295 | 0.436 | 0.43 | 0.4375 |

Table 2: Mean dissociation co-complexity values for all networks. Note that CNN and its variants maintain the dissociation co-complexity at a similar level to that of MLP.

## 4.2 Measuring Dissociation Co-Complexity

Smaller invariance co-complexity can be inconsequential, if the network's ability to discriminate between two different categories in $\mathcal{G}$ also suffers. Hence, we measure the dissociation co-complexity (in (8)) to check whether lower values of $R_m^I$ also affect their discriminatory potential. We compute $\mathcal{R}_m^D$ similarly to $\mathcal{R}_m^I$, by using 1000 randomly initialized networks (for the supremum in (8)), and then averaging the measure for 100 batches of data ($S$ and $S'$). Table 2 shows the results, where we find that the values of $R_m^D$ for all studied networks are very close to each other. Therefore, only the invariance co-complexity of these architectures will majorly decide the generalization error bounds in Theorem 6. This shows that, in spite of better invariance capabilities of CNNs (lower $\mathcal{R}_m^I$) and its transformation-equivariant extensions, those networks preserve discriminatory potential necessary for differentiating images that belong to different categories.

## 5 Reflections

The proposed co-complexity measures lead to an extended perspective on generalization error by accounting for ground truth generator spaces. The objective of introducing the generator space is to consider the structure in the ground truth labelling functions. New error bounds are proposed which consider various aspects of interaction between generator and classifier function spaces.

Co-complexity can be decomposed into two separate components, invariance co-complexity ($\mathcal{R}_m^I$) and dissociated co-complexity ($\mathcal{R}_m^D$), which are found to measure the degree to which the classifier's function space obeys the invariance transformations in the data ($\mathcal{R}_m^I$), and also the degree to which the classifier is able to differentiate between separate categories in the data ($\mathcal{R}_m^D$). If we interpret $\mathcal{R}_m(\mathcal{F}, \mathcal{G})$ as the *variance* of the classifier (see Theorems 3 and 4) and the training error as the *bias*, we see that $\mathcal{R}_m^I$ and $\mathcal{R}_m^D$ affect the training error (bias) and generalization gap (variance) differently (see Theorem 6). Theorem 6 outlines a clear objective for reducing variance while maintaining low bias: reduce $\mathcal{R}_m^I$ while keeping $\mathcal{R}_m^D$ and $R_m^{D,1}$ unchanged. We note that monitoring $\mathcal{R}_m^I$ and $\mathcal{R}_m^D$ can be useful for finding better learning architectures for a specific problem. Furthermore, it is also clear that classification problems which contain more invariance constraints, $\mathcal{R}_m^I$ plays a greater role in controlling the generalization gap (as $\alpha$ will be higher).

Experiments on MNIST and STL-10 reveal that the invariance co-complexity $\mathcal{R}_m^I$ for CNNs and their transformation-equivariant extensions is always lower than MLPs, while their dissociation co-complexities, $\mathcal{R}_m^D$ and $R_m^{D,1}$, are comparable to MLPs (see Appendix B.1 for experiments which measure $R_m^{D,1}$). Furthermore, transformation-equivariant versions of CNN show lower $\mathcal{R}_m^I$ than the vanilla CNN itself, while maintaining $\mathcal{R}_m^D$, i.e., they have low variance and low bias. Appendix B.3 presents results when neural networks of varying complexity are used as generator spaces and verifies that higher complexity generator spaces lead to a larger generalization gap.

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

## A   APPENDIX - TIGHTENING EXISTING RESULTS

For the following results, we primarily work in the setting where the ground truth generator space only contains a single element, which pertains to the conventional treatment of generalization error bounds with Rademacher complexity. Note that in this setting, we can still define the invariance classes of $\mathcal{G}$, which we will make use of. Furthermore, we primarily work with the correlated Rademacher complexity $\mathcal{R}_m^C(\mathcal{F})$, which was defined in section 2. We note that $\mathcal{R}_m^C(\mathcal{F})$ involves two instantiations of the dataset in $S$ and $S'$. Furthermore, we also observe that the final $\mathcal{R}_m^C(\mathcal{F})$ depends on how each datapoint $z_i$ in $S$ is paired with the corresponding datapoint $z_i'$ in $S'$,

$$\mathcal{R}_m^C(\mathcal{F}) = \frac{1}{2} \times \mathop{\mathbb{E}}_{\sigma, S, S'} \left[ \sup_{f \in \mathcal{F}} \left( \frac{1}{m} \sum_{i=1}^m f(z_i) f(z_i') \sigma_i \right) \right] \tag{19}$$

For the following results, we define the *invariance-matched* correlated Rademacher complexity $\mathcal{R}_m^{C_{inv}}(\mathcal{F})$, which chooses a pairing $(i, M(i))$ between the points in $S$ and $S'$, such that

$$M = \arg\max_M \sum_i |z_{M(i)}' \cap \mathcal{I}_{\mathcal{G}}(z_i)|. \tag{20}$$

That is, the matching function $M$ is chosen such that there exists the largest number of pairings $z_i, z_{M(i)}'$ where $z_{M(i)}'$ is present the invariance extended set of $z_i$. Thus, the *invariance-matched* correlated complexity can then be defined as

$$\mathcal{R}_m^{C_{inv}}(\mathcal{F}) = \frac{1}{2} \times \mathop{\mathbb{E}}_{\sigma, S, S'} \left[ \sup_{f \in \mathcal{F}} \left( \frac{1}{m} \sum_{i=1}^m f(z_i) f(z_{M(i)}') \sigma_i \right) \right], \tag{21}$$

where $M$ is chosen using equation 20. Given this, we have the following theorem.

**Theorem 7.** *Consider the classifier function space $\mathcal{F}$, and the previous defined correlated Rademacher complexity of $\mathcal{F}$, $\mathcal{R}_m^C(\mathcal{F})$. Furthermore, let us assume that the generator space only contains a single element, i.e., the conventional setting ($\mathcal{R}_m(\mathcal{G}) = 0$). Then we have for any $f \in \mathcal{F}$, with a probability $p \geq 1 - \delta$, we have*

$$err_P(f) \leq \widehat{err}_S(f) + \mathcal{R}_m^{C_{inv}}(\mathcal{F}) + \sqrt{\frac{\log(1/\delta)}{2m}}. \tag{22}$$

*Proof.* First, we will show that for any $f \in \mathcal{F}$, with a probability $p \geq 1 - \delta$,

$$\text{err}_P(f) \leq \widehat{\text{err}}_S(f) + \mathcal{R}_m^C(\mathcal{F}) + \sqrt{\frac{\log(1/\delta)}{2m}}. \tag{23}$$

This result trivially follows from Theorem 3, when $\mathcal{G}$ contains a single element denoted by $g_0$, as then

$$\mathcal{R}_m(\mathcal{F}, \mathcal{G}) = \frac{1}{2} \times \mathop{\mathbb{E}}_{\sigma, S, S'} \left[ \sup_{f \in \mathcal{F}} \left( \frac{1}{m} \sum_{i=1}^{m} f(z_i) f(z_i') g_0(z_i) g_0(z_i') \sigma_i \right) \right] \tag{24}$$

$$= \frac{1}{2} \times \mathop{\mathbb{E}}_{\sigma, S, S'} \left[ \sup_{f \in \mathcal{F}} \left( \frac{1}{m} \sum_{i=1}^{m} f(z_i) f(z_i') \sigma_i \right) \right] \tag{25}$$

$$= \mathcal{R}_m^C(\mathcal{F}) \tag{26}$$

Note that as $\mathcal{R}_m^C(\mathcal{F}) \leq \mathcal{R}_m(\mathcal{F})$, this is a tighter global bound that the original Rademacher bound.

Then, the result immediately follows when we recognize that changing the ordering of $z_i$ and $z_i'$ does not affect the value of the generalization gap in (57). Furthermore, the matching $M$ for each sampled $S, S'$ is independent of the choice of the Rademacher variables $\sigma_i$, and thus the subsequent steps follow in the same way as in (67). Subsequently, we still have with a probability $p \geq 1 - \delta$,

$$\mathrm{err}_P(f) \leq \widehat{\mathrm{err}}_S(f) + \mathcal{R}_m^{C_{inv}}(\mathcal{F}) + \sqrt{\frac{\log(1/\delta)}{2m}}. \tag{27}$$

Note that when the invariance co-complexity of $\mathcal{F}$ w.r.t. $\mathcal{G}$ is smaller than its dissociation co-complexity, then the above is expected to be a tighter bound than (23). This is because the invariance-aware matching function $M$ leads to a greater number of pairings $z_i, z_i'$ where $z_i' \in \mathcal{I}_{\mathcal{G}}(z_i)$ (thus lies in its invariance extended set).

$\square$

## B  APPENDIX - ADDITIONAL EXPERIMENTS

### B.1  EXPERIMENT: MEASURING DISSOCIATION CO-COMPLEXITY ($\mathcal{R}_m^{D,1}(\mathcal{F}, \mathcal{G})$)

In the main paper, we measure the invariance co-complexities $\mathcal{R}_m^I(\mathcal{F}, \mathcal{G})$ and the dissociation co-complexities $\mathcal{R}_m^D(\mathcal{F}, \mathcal{G})$ of four different network types: Multi-layer Perceptron (MLP), Vanilla CNN (CNN), Scale-Equivariant CNN (SE-CNN in Sosnovik et al. (2019)) and Rotation Equivariant CNN (RE-CNN in Cohen & Welling (2016b)). In this experiment, we measure the single-sample dissociation co-complexity $\mathcal{R}_m^{D,1}(\mathcal{F}, \mathcal{G})$ of these networks, as $\mathcal{R}_m^{D,1}(\mathcal{F}, \mathcal{G})$ also affects the training error in the generalization error bound in Theorem 6. Please note that the datasets used in our experiments (MNIST and STL-10), have 10 categories. However, since the invariance and dissociation co-complexities can only be computed binary classifiers, we use random subsets of the data which contain only two categories, for computing the value of co-complexity.

Table 3 shows the results. We find that in both datasets, the $\mathcal{R}_m^{D,1}(\mathcal{F}, \mathcal{G})$ values of the tested networks are very close to each other. This is consistent with the dissociation co-complexity $\mathcal{R}_m^D(\mathcal{F}, \mathcal{G})$, shown in Table 2 of the main paper. Together with those results, it is clear that only the invariance co-complexities $\mathcal{R}_m^I(\mathcal{F}, \mathcal{G})$ will mainly contribute to the generalization error of these networks, as the dissociation co-complexities $\mathcal{R}_m^{D,1}(\mathcal{F}, \mathcal{G})$ and $\mathcal{R}_m^D(\mathcal{F}, \mathcal{G})$ do not show any significant variation.

For example, from Table 1 and 2 of the main paper, we see that the invariance co-complexity (rotation) of the RE-CNN (0.357) is smaller than that of the MLP (0.424), whereas the $\mathcal{R}_m^{D,1}(\mathcal{F}, \mathcal{G})$ (0.4492 vs. 0.4472) and $\mathcal{R}_m^D(\mathcal{F}, \mathcal{G})$ (0.438 vs. 0.4415) of these two networks do not show much variation.

|  | MLP | CNN | SE-CNN | RE-CNN |
|---|---|---|---|---|
| Dissociation Co-Complexity ($\mathcal{R}_m^{D,1}(\mathcal{F}, \mathcal{G})$, MNIST) | 0.4492 | 0.4589 | 0.4479 | 0.4472 |
| Dissociation Co-Complexity ($\mathcal{R}_m^{D,1}(\mathcal{F}, \mathcal{G})$, STL-10) | 0.4352 | 0.4465 | 0.4405 | 0.4570 |

Table 3: Mean dissociation co-complexity values for all networks. Note that CNN and its variants maintain the dissociation co-complexity at a similar level to that of MLP.

### B.2 EXPERIMENT: MEASURING INVARIANCE CO-COMPLEXITY FOR DIFFERENT TRANSFORMATION PARAMETER CHOICES

In addition to the experiments which estimate the overall invariance co-complexity of the various network baselines, we also showcase how $R_m^I$ depends on the exact transformation parameter $t$ for each case, in Fig. 3. Note that equivariant networks usually have low $R_m^I$ for most values of $t$, except for the RE-CNN which is tuned to be invariant to rotations of $90°$, and therefore only showcases significant drops in $R_m^I$ when $t$ is near $90°$ or $180°$.

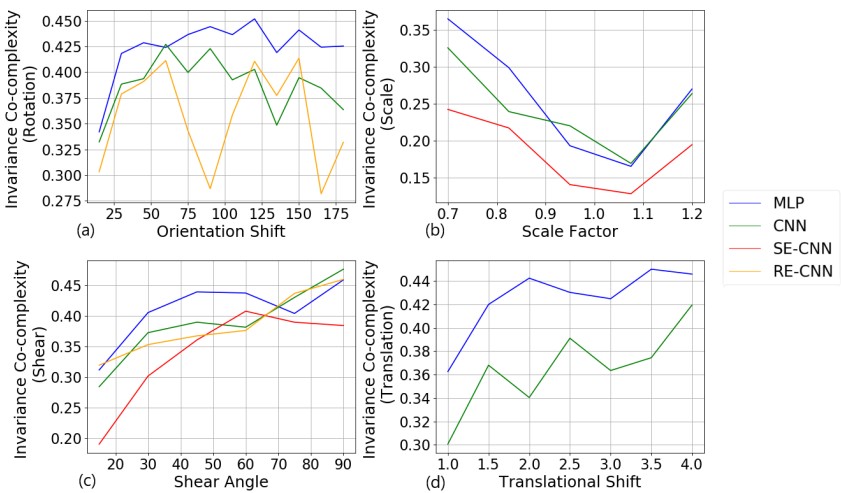

Figure 2: Invariance co-complexity values for the relevant networks on MNIST, when the datapoint pairs $S$ and $S'$ are separated by varying degrees of rotational, scale, shear and translational shifts.

### B.3 EXPERIMENT: TESTING GENERALIZATION ERROR BOUNDS WITH NEURAL NETWORK GENERATOR SPACES

We consider a setting where $\mathcal{F}$ is the space of linear classifiers and the generator space $\mathcal{G}$ is the space of relu-activated neural networks with a single hidden layer containing $H$ hidden neurons. We have $\mathcal{R}_m(\mathcal{F}) \propto \sqrt{1/m}$ and $\mathcal{R}_m(\mathcal{G}) \propto \sqrt{H/m}$. Closed form expressions for co-complexity $\mathcal{R}_m(\mathcal{F}, \mathcal{G})$ are non-trivial to compute, but we use the property $\mathcal{R}_m(\mathcal{F}, \mathcal{G}) \leq \mathcal{R}_m(\mathcal{F}) + \mathcal{R}_m(\mathcal{G})$ (Theorem 2). Theorem 2 implies that only increasing the complexity of $\mathcal{G}$ (by increasing $H$), should lead to a steady increase in the difference between training and testing error (i.e. the generalization gap). We execute this scenario with two variations: $m = 10$ and $m = 100$. $H$ was chosen from $\{2, 4, 8, 16, 32\}$. Note that the data distribution is kept fixed for all $H$. Therefore, in this experiment, only the label generating function is changed.

We observe that the expected generalization gap steadily increases (ranged between 0.6%-8.9%) as we increase the complexity of the generator class (for both $m = 10$ and $m = 100$). Furthermore, we find a very high degree of correlation (0.982 for $m = 10$ and 0.976 for $m = 100$) between the generalization gap trend predicted by Theorem 2 ($\approx \sqrt{H/m}$) and the empirically observed average generalization gap. This shows that the complexity of the generator class also directly controls the generalization gap, when everything else is unchanged. This result clearly highlights that when additional information about the generator space is known, more accurate trends of generalization error can be estimated by using both $\mathcal{R}_m(\mathcal{G})$ and $\mathcal{R}_m(\mathcal{F})$.

## C   APPENDIX - PROOF OF THEORETICAL RESULTS

In what follows, we provide the proofs of Theorems 2 to 6. The proof of Theorem 1 of the main paper is available in Bartlett & Mendelson (2003). We also prove, in Lemma 2, the properties of co-complexity stated in Section 5 of the main paper. We use the definitions provided in Section 5 of the main paper.

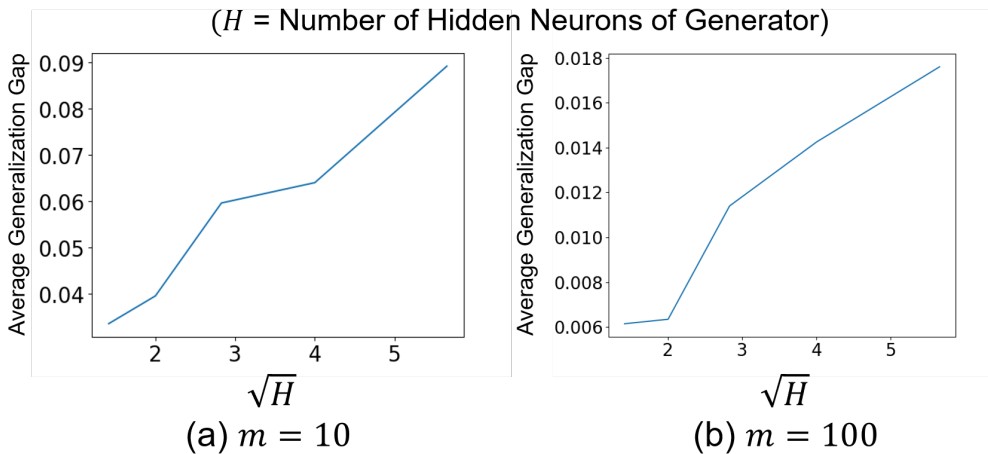

Figure 3: Plots depicting the average generalization gap (difference between test and training error) when 2-layer neural networks of variable hidden neuron number $H$, are chosen as the label generating function, and a single layer neural network is chosen as the classifier. The generalization gap trend is plotted for two scenarios, (a) $m = 10$ and (b) $m = 100$, where $m$ is the number of training examples. As here $\mathcal{R}_m(\mathcal{F})$ is fixed, but $\mathcal{R}_m(\mathcal{G}) \propto \sqrt{H}$ changes with $H$, we plot the generalization gap averages against $\sqrt{H}$. Note that the trend is linear in nature in both cases (correlation of $\approx 0.98$ for both), showing that the generalization gap depends directly on the complexity of the generator space.

Please note that for the following proofs, a variable $z$ which follows the Rademacher distribution ($Pr(z = +1) = Pr(z = -1) = 0.5$) are referred to as Rademacher variable.

### C.1 Proofs of Properties **P1-P3** of Co-complexity

We begin with the proof of the properties of co-complexity (**P1-P3**) and a property of the correlated Rademacher complexity, which are described in Section 5 of the main paper.

**Lemma 2.** *For any two function spaces $\mathcal{F}$ and $\mathcal{G}$, the following statements hold.*

$$\mathcal{R}_m(\mathcal{F}, \mathcal{G}) = \mathcal{R}_m(\mathcal{G}, \mathcal{F}) \tag{28}$$

$$\mathcal{R}_m(\mathcal{F}, \mathcal{G}) \geq \mathcal{R}_m^C(\mathcal{F}) \tag{29}$$

$$\mathcal{R}_m(\mathcal{F}, \mathcal{G}) \geq \mathcal{R}_m^C(\mathcal{G}) \tag{30}$$

$$\mathcal{R}_m(\mathcal{F}, \mathcal{G}) \leq \mathcal{R}_m^C(\mathcal{F}) + \mathcal{R}_m^C(\mathcal{G}) \tag{31}$$

$$\mathcal{R}_m^C(\mathcal{F}) \leq \mathcal{R}_m(\mathcal{F}) \tag{32}$$

*Proof.* (i) To prove the first statement, it is trivial to see that interchanging $f$ and $g$ in the expression of $\mathcal{R}_m(\mathcal{F}, \mathcal{G})$ keeps it unchanged.

Thus, $\mathcal{R}_m(\mathcal{F}, \mathcal{G}) = \mathcal{R}_m(\mathcal{G}, \mathcal{F})$.

(ii) Let us define $\sigma_i' = g_0(z_i)g_0(z_i')\sigma_i$. To prove the second statement, we choose any fixed function $g_0$ within $\mathcal{G}$ and we have,

$$\frac{1}{2} \times \mathop{\mathbb{E}}_{\sigma,S,S'} \left[ \sup_{f\in\mathcal{F},g\in\mathcal{G}} \left( \frac{1}{m}\sum_{i=1}^{m} f(z_i)f(z_i')g(z_i)g(z_i')\sigma_i \right) \right]$$

$$\geq \frac{1}{2} \times \mathop{\mathbb{E}}_{\sigma,S,S'} \left[ \sup_{f\in\mathcal{F}} \left( \frac{1}{m}\sum_{i=1}^{m} g_0(z_i)g_0(z_i')f(z_i)f(z_i')\sigma_i \right) \right] \tag{33}$$

$$= \frac{1}{2} \times \mathop{\mathbb{E}}_{\sigma',S,S'} \left[ \sup_{f\in\mathcal{F}} \left( \frac{1}{m}\sum_{i=1}^{m} f(z_i)f(z_i')\sigma_i' \right) \right] \tag{34}$$

$$= \mathcal{R}_m^C(\mathcal{F}). \tag{35}$$

Here, we use the fact that $\sigma_i'$ is also a Rademacher variable.

(iii) We can prove the third statement in the same manner, fixing $f_0 \in \mathcal{F}$ instead.

(iv) The fourth statement can be proven as follows. First, we note that $f(z_i)f(z_i')$ and $g(z_i)g(z_i')$ are variables which take the value of 1 or $-1$. Thus, $f(z_i)f(z_i')g(z_i)g(z_i')$ can be expressed as $(1 - |f(z_i)f(z_i') - g(z_i)g(z_i')|) = (1 - f(z_i)f(z_i') - g(z_i)g(z_i')v_i)$, where $v_i \in \{-1, -1\}$ depends on $f(z_i)f(z_i')$ and $g(z_i)g(z_i')$ together, but is independent of each of them individually. We can therefore write,

$$\frac{1}{2} \times \mathop{\mathbb{E}}_{\sigma,S,S'} \left[ \sup_{f\in\mathcal{F},g\in\mathcal{G}} \left( \frac{1}{m}\sum_{i=1}^{m} f(z_i)f(z_i')g(z_i)g(z_i')\sigma_i \right) \right]$$

$$= \frac{1}{2} \times \mathop{\mathbb{E}}_{\sigma,S,S'} \left[ \sup_{f\in\mathcal{F},g\in\mathcal{G}} \left( \frac{1}{m}\sum_{i=1}^{m} \left(1 - |f(z_i)f(z_i') - g(z_i)g(z_i')|\right)\sigma_i \right) \right] \tag{36}$$

$$= \frac{1}{2} \times \mathop{\mathbb{E}}_{\sigma,S,S'} \left[ \sup_{f\in\mathcal{F},g\in\mathcal{G}} \left( \frac{1}{m}\sum_{i=1}^{m} \left(f(z_i)f(z_i') - g(z_i)g(z_i')\right)v_i\sigma_i \right) \right]. \tag{37}$$

Now, consider any fixed $\sigma = \{\sigma_0, ..., \sigma_m\} = \{\sigma_0^0, ..., \sigma_m^0\}$, and fix $S$ and $S'$. First, let us denote

$$f_{opt}, g_{opt} = \arg\max_{f\in\mathcal{F},g\in\mathcal{G}} \left[ \frac{1}{m}\sum_{i=1}^{m} \left(f(z_i)f(z_i') - g(z_i)g(z_i')\right)v_i\sigma_i^0 \right] \tag{38}$$

Let us denote the final $v_i$ for $f_{opt}$ and $g_{opt}$ by $v_i^{opt}$. Note that $v_i^{opt}$ here depends on $f_{opt}(z_i)f_{opt}(z_i')$ and $g_{opt}(z_i)g_{opt}(z_i')$. Note that $v_1^{opt}, v_2^{opt}, ..v_m^{opt}$ will be independent of each other, as $\sigma_1, \sigma_2, ..\sigma_m$ are independent. Also observe that $v_1^{opt}, v_2^{opt}, ..v_m^{opt}$ will be independent of $\sigma_1^0, \sigma_2^0, ..\sigma_m^0$ respectively, as they only depend on $f_{opt}$ and $g_{opt}$. We define $\sigma_i' = v_i^{opt}\sigma_i^0$. We note that $\sigma_1', \sigma_2', ..\sigma_m'$ will be independent across all samples, and $Pr(\sigma_i' = +1) = Pr(\sigma_i' = -1) = 0.5$, same as $\sigma_i$. Thus $\sigma_1', \sigma_2', ..\sigma_m'$ are i.i.d Rademacher variables like $\sigma_1, \sigma_2, ..\sigma_m$. We thus note,

$$\sup_{f\in\mathcal{F},g\in\mathcal{G}} \left( \frac{1}{m}\sum_{i=1}^{m} (f(z_i)f(z_i') - g(z_i)g(z_i'))v_i\sigma_i^0 \right)$$

$$= \frac{1}{m}\sum_{i=1}^{m} \left(f_{opt}(z_i)f_{opt}(z_i') - g_{opt}(z_i)g_{opt}(z_i')\right)v_i^{opt}\sigma_i^0 \tag{39}$$

$$= \frac{1}{m}\sum_{i=1}^{m} \left(f_{opt}(z_i)f_{opt}(z_i') - g_{opt}(z_i)g_{opt}(z_i')\right)\sigma_i' \tag{40}$$

$$\leq \sup_{f\in\mathcal{F},g\in\mathcal{G}} \left( \frac{1}{m}\sum_{i=1}^{m} (f(z_i)f(z_i') - g(z_i)g(z_i'))\sigma_i' \right), \tag{41}$$

where in the final supremum $\sigma_i'$ are kept fixed at their original values ($\sigma_i' = v_i^{opt}\sigma_i^0$). We then have,

$$\frac{1}{2} \times \mathop{\mathbb{E}}_{\sigma,S,S'} \left[ \sup_{f \in \mathcal{F}, g \in \mathcal{G}} \left( \frac{1}{m} \sum_{i=1}^{m} f(z_i)f(z_i')g(z_i)g(z_i')\sigma_i \right) \right]$$

$$\leq \frac{1}{2} \times \mathop{\mathbb{E}}_{\sigma',S,S'} \left[ \sup_{f \in \mathcal{F}, g \in \mathcal{G}} \left( \frac{1}{m} \sum_{i=1}^{m} \left( f(z_i)f(z_i') - g(z_i)g(z_i') \right) \sigma_i' \right) \right], \tag{42}$$

$$= \mathcal{R}_m^C(\mathcal{F}) + \mathcal{R}_m^C(\mathcal{G}). \tag{43}$$

The final result $\mathcal{R}_m^C(\mathcal{F}) \leq \mathcal{R}_m(\mathcal{F})$ can be shown as follows. To prove this, we use the fact that $f(z_i)f(z_i') = 1 - |f(z_i) - f(z_i')|$. We have,

$$\mathcal{R}_m^C(\mathcal{F}) = \frac{1}{2} \times \mathop{\mathbb{E}}_{\sigma,S,S'} \left[ \sup_{f \in \mathcal{F}} \left( \frac{1}{m} \sum_{i=1}^{m} f(z_i)f(z_i')\sigma_i \right) \right] \tag{44}$$

$$= \frac{1}{2} \times \mathop{\mathbb{E}}_{\sigma,S,S'} \left[ \sup_{f \in \mathcal{F}} \left( \frac{1}{m} \sum_{i=1}^{m} (1 - |f(z_i) - f(z_i')|)\sigma_i \right) \right] \tag{45}$$

$$= \frac{1}{2} \times \mathop{\mathbb{E}}_{\sigma,S,S'} \left[ \sup_{f \in \mathcal{F}} \left( \frac{1}{m} \sum_{i=1}^{m} (f(z_i) - f(z_i'))v_i\sigma_i \right) \right] \tag{46}$$

Here, $v_i$ takes values in $\{-1, 1\}$, depending on whether $f(z_i)$ or $f(z_i')$ is greater. Note that as $\sigma_i$ is a Rademacher variable, $v_i\sigma_i$ is also a Rademacher variable. We also note that $v_i$ is independent of $f(z_i)$ and $f(z_i')$ individually. Thus we have,

$$\mathcal{R}_m^C(\mathcal{F}) = \frac{1}{2} \times \mathop{\mathbb{E}}_{\sigma,S,S'} \left[ \sup_{f \in \mathcal{F}} \left( \frac{1}{m} \sum_{i=1}^{m} (f(z_i) - f(z_i'))v_i\sigma_i \right) \right] \tag{47}$$

$$\leq \frac{1}{2} \times \mathop{\mathbb{E}}_{\sigma',S} \left[ \sup_{f \in \mathcal{F}} \left( \frac{1}{m} \sum_{i=1}^{m} f(z_i)(\sigma_i') \right) \right] + \frac{1}{2} \times \mathop{\mathbb{E}}_{\sigma',S'} \left[ \sup_{f \in \mathcal{F}} \left( \frac{1}{m} \sum_{i=1}^{m} f(z_i')(-\sigma_i') \right) \right], \tag{48}$$

where $\sigma_i' = v_i\sigma_i$. As $-\sigma_i'$ is also a Rademacher variable, we finally have

$$\mathcal{R}_m^C(\mathcal{F}) \leq \frac{1}{2} \times \mathop{\mathbb{E}}_{\sigma',S} \left[ \sup_{f \in \mathcal{F}} \left( \frac{1}{m} \sum_{i=1}^{m} f(z_i)(\sigma_i') \right) \right] + \frac{1}{2} \times \mathop{\mathbb{E}}_{\sigma',S'} \left[ \sup_{f \in \mathcal{F}} \left( \frac{1}{m} \sum_{i=1}^{m} f(z_i')(\sigma_i') \right) \right] \tag{49}$$

$$= \frac{\mathcal{R}_m(\mathcal{F}) + \mathcal{R}_m(\mathcal{F})}{2} \tag{50}$$

$$= \mathcal{R}_m(\mathcal{F}). \tag{51}$$

This completes the proofs. □

### C.2 PROOF OF THEOREM 2

The following result proves a generalization error bound which incorporates $\mathcal{R}_m(\mathcal{F})$ and $\mathcal{R}_m(\mathcal{G})$.

**Theorem 2.** *For $0 < \delta < 1$, with probability $p \geq 1 - \delta$, we have*

$$err_P(f) \leq \widehat{err}_S(f) + \mathcal{R}_m(\mathcal{F}) + \mathcal{R}_m(\mathcal{G}) + \sqrt{\frac{\log(1/\delta)}{2m}}. \tag{52}$$

*Proof.* First, we reiterate that $\widehat{err}_S(f) = \sum_{i=1}^{m} \left( 1 - f(z_i)g_{truth}(z_i) \right) / 2m$, for some $g_{truth} \in \mathcal{G}$, and similarly for $err_P(f)$, where $m \to \infty$.

Then, we retrace the steps of the original proof in Bartlett & Mendelson (2003), noting that

$$\mathrm{err}_P(f) \le \widehat{\mathrm{err}}_S(f) + \sup_{f \in \mathcal{F}, g \in \mathcal{G}, g(S) = g_{truth}(S)} (\mathrm{err}_P(f) - \widehat{\mathrm{err}}_S(f)) \tag{53}$$

$$\le \widehat{\mathrm{err}}_S(f) + \sup_{f \in \mathcal{F}, g \in \mathcal{G}} (\mathrm{err}_P(f) - \widehat{\mathrm{err}}_S(f)). \tag{54}$$

Here we initially only consider functions $g$ in $\mathcal{G}$ such that $g(z_0) = g_{truth}(z_0), g(z_1) = g_{truth}(z_1), ..., g(z_m) = g_{truth}(z_m)$. Notice, that differently from (Bartlett & Mendelson (2003)), the supremum eventually considers the generator space $\mathcal{G}$ in addition to the classifier's function space $\mathcal{F}$. We define $\phi(S) = \sup_{f \in \mathcal{F}, g \in \mathcal{G}} (\mathrm{err}_P(f) - \widehat{\mathrm{err}}_S(f))$, and notice that

$$\sup_{z_1, z_2, ..., z_m, z_i' \in \mathcal{Z}} |\phi(z_1, z_2, ..., z_i, ...., z_m) - \phi(z_1, z_2, ..., z_i', ...., z_m)| \le \frac{1}{m}. \tag{55}$$

This leads to the application of McDiarmid's Inequality (McDiarmid (1989)) on the concentration bounds for $\phi(S)$, which results in the the following bound, with a probability of at least $1 - \delta$,

$$\mathrm{err}_P(f) \le \widehat{\mathrm{err}}_S(f) + \mathbb{E}_S \left[ \sup_{f \in \mathcal{F}, g \in \mathcal{G}} (\mathrm{err}_P(f) - \widehat{\mathrm{err}}_S(f)) \right] + \sqrt{\frac{\log(1/\delta)}{m}}. \tag{56}$$

Finally, we express $\mathbb{E}_S[\phi(S)]$ as follows.

$$\mathbb{E}_S \left[ \sup_{f \in \mathcal{F}, g \in \mathcal{G}} (\mathrm{err}_P(f) - \widehat{\mathrm{err}}_S(f)) \right] = \mathbb{E}_S \left[ \sup_{f \in \mathcal{F}, g \in \mathcal{G}} \mathbb{E}_{S'} [\widehat{\mathrm{err}}_{S'}(f) - \widehat{\mathrm{err}}_S(f)|S] \right] \tag{57}$$

$$= \mathbb{E}_S \left[ \sup_{f \in \mathcal{F}, g \in \mathcal{G}} \mathbb{E}_{S'} \left[ \frac{1}{m} \sum_{i=1}^{m} \left( \frac{1 - f(z_i')g(z_i')}{2} - \frac{1 - f(z_i)g(z_i)}{2} \right) |S| \right] \right] \tag{58}$$

$$\le \mathbb{E}_{S,S'} \left[ \sup_{f \in \mathcal{F}, g \in \mathcal{G}} \left( \frac{1}{m} \sum_{i=1}^{m} \frac{f(z_i)g(z_i) - f(z_i')g(z_i')}{2} \right) \right]. \tag{59}$$

The last inequality (Jensen's inequality (Cover & Thomas (2012))) is applicable because sup is a convex function. We proceed by multiplying the terms with Rademacher variables $\sigma_i$, which randomly take on $\{-1, 1\}$ with equal probability. This keeps the expected value unchanged. Note that $\mathbb{E}[\sigma_i] = 0$. In what follows, we express $f(z_i)g(z_i) = 1 - |f(z_i) - g(z_i)|$. This is enabled because both have their range in $\{-1, 1\}$.

$$\mathbb{E}_{S,S'} \left[ \sup_{f \in \mathcal{F}, g \in \mathcal{G}} \left( \frac{1}{m} \sum_{i=1}^{m} \frac{f(z_i)g(z_i) - f(z_i')g(z_i')}{2} \right) \right]$$

$$= \mathbb{E}_{S,S',\sigma} \left[ \sup_{f \in \mathcal{F}, g \in \mathcal{G}} \left( \frac{1}{m} \sum_{i=1}^{m} \frac{(f(z_i)g(z_i) - f(z_i')g(z_i'))\sigma_i}{2} \right) \right] \tag{60}$$

$$\le \mathbb{E}_{\sigma,S,S'} \left[ \sup_{f \in \mathcal{F}, g \in \mathcal{G}} \left( \frac{1}{m} \sum_{i=1}^{m} \frac{|f(z_i) - g(z_i)|\sigma_i}{2} \right) \right]$$

$$+ \mathbb{E}_{\sigma,S,S'} \left[ \sup_{f \in \mathcal{F}, g \in \mathcal{G}} \left( \frac{1}{m} \sum_{i=1}^{m} \frac{|f(z_i') - g(z_i')|\sigma_i}{2} \right) \right]. \tag{61}$$

Note that multiplication with the Rademacher variables does not change the value of the expectation, as the Rademacher variable $\sigma_i$ simply controls whether $z_i$ and $z_i'$ belong to $S$ and $S'$ ($\sigma_i = 1$), or to $S'$ and $S'$ ($\sigma_i = -1$). Coupled with the fact that $\mathbb{E}[\sigma_i] = 0$, we note that this keeps the expectation in (59) unchanged. Now, note that $(|f(z_i) - g(z_i)|)\sigma_i = (f(z_i) - g(z_i))v_i\sigma_i$. Here $v_i$ is also a

variable that takes its values in $\{-1, 1\}$. Since $\sigma_i$ is a Rademacher variable, $v_i\sigma_i = \sigma'_i$ is also a Rademacher variable. In what follows, we use the fact that $v_i$ is independent of $f_i$ and $g_i$ individually.

$$
\mathop{\mathbb{E}}_{\sigma,S,S'} \left[ \sup_{f\in\mathcal{F},g\in\mathcal{G}} \left( \frac{1}{m} \sum_{i=1}^{m} \frac{|f(z_i) - g(z_i)|\sigma_i}{2} \right) \right] \leq \mathop{\mathbb{E}}_{\sigma',S,S'} \left[ \sup_{f\in\mathcal{F}} \left( \frac{1}{m} \sum_{i=1}^{m} \frac{f(z_i)\sigma'_i}{2} \right) \right]
$$

$$
+ \mathop{\mathbb{E}}_{\sigma',S,S'} \left[ \sup_{g\in\mathcal{G}} \left( \frac{1}{m} \sum_{i=1}^{m} \frac{g(z_i)\sigma'_i}{2} \right) \right] \quad (62)
$$

$$
= \frac{\mathcal{R}_m(\mathcal{F}) + \mathcal{R}_m(\mathcal{G})}{2}. \quad (63)
$$

Similarly, we can derive the same for $\mathbb{E}_{\sigma,S,S'} \left[ \sup_{f\in\mathcal{F},g\in\mathcal{G}} \left( \frac{1}{m} \sum_{i=1}^{m} \frac{|f(z'_i) - g(z'_i)|\sigma_i}{2} \right) \right]$. Thus, finally, we can represent the upper bound on $\mathbb{E}[\phi(S)]$ as follows .

$$
\mathbb{E}[\phi(S)] \leq \frac{\mathcal{R}_m(\mathcal{F}) + \mathcal{R}_m(\mathcal{G})}{2} + \frac{\mathcal{R}_m(\mathcal{F}) + \mathcal{R}_m(\mathcal{G})}{2} = \mathcal{R}_m(\mathcal{F}) + \mathcal{R}_m(\mathcal{G}). \quad (64)
$$

This leads to the final form of the generalization error bound. With probability at least $1 - \delta$, we have

$$
\text{err}_P(f) \leq \widehat{\text{err}}_S(f) + \mathcal{R}_m(\mathcal{F}) + \mathcal{R}_m(\mathcal{G}) + \sqrt{\frac{\log(1/\delta)}{m}}. \quad (65)
$$

$\square$

### C.3 Proof of Theorem 3

The following statements propose the generalization error bound in terms of co-complexity.

**Theorem 3.** *For $0 < \delta < 1$, with probability $p \geq 1 - \delta$, we have*

$$
err_P(f) \leq \widehat{err}_S(f) + \mathcal{R}_m(\mathcal{F}, \mathcal{G}) + \sqrt{\frac{\log(1/\delta)}{2m}}. \quad (66)
$$

*Proof.* We proceed in the same way as in appendix C.2, until we arrive at (57). In what follows, we note that we can write $f(z_i)g(z_i) - f(z'_i)g(z'_i) = (1 - f(z_i)g(z_i)f(z'_i)g(z'_i))v_i$, for some binary valued variable $v_i$ with range in $\{-1, 1\}$. We also multiply the original Rademacher variable $\sigma_i$ to (57), and derive the following set of bounds.

$$
\mathop{\mathbb{E}}_S \left[ \sup_{f\in\mathcal{F},g\in\mathcal{G}} (\text{err}_P(f) - \widehat{\text{err}}_S(f)) \right] \leq \mathop{\mathbb{E}}_{S,S'} \left[ \sup_{f\in\mathcal{F},g\in\mathcal{G}} \left( \frac{1}{m} \sum_{i=1}^{m} \frac{f(z_i)g(z_i) - f(z'_i)g(z'_i)}{2} \right) \right] \quad (67)
$$

$$
= \mathop{\mathbb{E}}_{S,S'} \left[ \sup_{f\in\mathcal{F},g\in\mathcal{G}} \left( \frac{1}{m} \sum_{i=1}^{m} \frac{(1 - f(z_i)g(z_i)f(z'_i)g(z'_i))\,v_i}{2} \right) \right] \quad (68)
$$

$$
= \mathop{\mathbb{E}}_{\sigma,S,S'} \left[ \sup_{f\in\mathcal{F},g\in\mathcal{G}} \left( \frac{1}{m} \sum_{i=1}^{m} \frac{(1 - f(z_i)g(z_i)f(z'_i)g(z'_i))\,v_i\sigma_i}{2} \right) \right] \quad (69)
$$

$$
\leq \mathop{\mathbb{E}}_{\sigma',S,S'} \left[ \sup_{f\in\mathcal{F},g\in\mathcal{G}} \left( \frac{1}{m} \sum_{i=1}^{m} \frac{(f(z_i)g(z_i)f(z'_i)g(z'_i))\,\sigma'_i}{2} \right) \right]. \quad (70)
$$

The last step follows from the fact that $\mathbb{E}[\sigma_i] = 0$, and the assumption that $\sigma'_i = v_i\sigma_i$, are also i.i.d Rademacher variables. This assumption will hold for most well behaved classifiers, for which less than half of the terms in (68) are greater than zero (which is applicable for classifiers with not too

large $\mathcal{R}_m(\mathcal{F})$). As $\frac{1}{2} \times \mathbb{E}_{\sigma', S, S'} \left[ \sup_{f \in \mathcal{F}, g \in \mathcal{G}} \left( \frac{1}{m} \sum_{i=1}^m f(z_i) g(z_i) f(z_i') g(z_i') \sigma_i' \right) \right] = \mathcal{R}_m(\mathcal{F}, \mathcal{G})$, we have the final generalization bound as follows. With probability $\geq 1 - \delta$,

$$\mathrm{err}_P(f) \leq \widehat{\mathrm{err}}_S(f) + \mathcal{R}_m(\mathcal{F}, \mathcal{G}) + \sqrt{\frac{\log(1/\delta)}{m}}. \tag{71}$$

$\square$

The above theorem leads to the following corollary.

**Corollary 3.1.** *We consider the hypothetical case where the roles are reversed, i.e., when the generator function space $\mathcal{G}$ is used to fit the labels, which are now generated by a function $f \in \mathcal{F}$. For any function $g \in \mathcal{G}$, for $0 < \delta < 1$, with probability $p \geq 1 - \delta$, we have*

$$err_P(g) \leq \widehat{err}_S(g) + \mathcal{R}_m(\mathcal{F}, \mathcal{G}) + \sqrt{\frac{\log(1/\delta)}{2m}}. \tag{72}$$

*Proof.* This is a direct outcome of the fact that $\mathcal{R}_m(\mathcal{F}, \mathcal{G}) = \mathcal{R}_m(\mathcal{G}, \mathcal{F})$. When the data labels in $S$ is generated by functions in $\mathcal{F}$ rather than $\mathcal{G}$, we will simply have with probability $p \geq 1 - \delta$,

$$\mathrm{err}_P(g) \leq \widehat{\mathrm{err}}_S(g) + \mathcal{R}_m(\mathcal{G}, \mathcal{F}) + \sqrt{\frac{\log\left(\frac{1}{\delta}\right)}{m}} \tag{73}$$

$$= \widehat{\mathrm{err}}_S(g) + \mathcal{R}_m(\mathcal{F}, \mathcal{G}) + \sqrt{\frac{\log\left(\frac{1}{\delta}\right)}{m}}. \tag{74}$$

Note that here the labels in $S$ are generated by a certain $f \in \mathcal{F}$. This completes the proof. $\square$

### C.4 PROOF OF THEOREM 4

The following result outlines the lower bound on the error function.

**Theorem 4.** *For $0 < \delta < 1$, with probability $p \geq 1 - \delta$,*

$$err_P(f) \geq \widehat{err}_S(f) - \mathcal{R}_m(\mathcal{F}, \mathcal{G}) - \sqrt{\frac{\log(1/\delta)}{2m}}. \tag{75}$$

*Proof.* First, we note that

$$\mathrm{err}_P(f) \geq \widehat{\mathrm{err}}_S(f) - \mathbb{E}_S \left[ \sup_{f \in \mathcal{F}, g \in \mathcal{G}} (\widehat{\mathrm{err}}_S(f) - \mathrm{err}_P(f)) \right] \tag{76}$$

Next, we denote

$$\phi(S) = \sup_{f \in \mathcal{F}, g \in \mathcal{G}} (\mathrm{err}_S(f) - \widehat{\mathrm{err}}_P(f)). \tag{77}$$

We then proceed similarly to Theorem 2 and Theorem 3, and we can show that

$$\mathbb{E}_S [\phi(S)] \leq \mathcal{R}_m(\mathcal{F}, \mathcal{G}). \tag{78}$$

Furthermore, note that

$$\sup_{z_1, z_2, ..., z_n, z_i' \in \mathcal{Z}} |\phi(z_1, z_2, ..., z_i, ...., z_m) - \phi(z_1, z_2, ..., z_i', ...., z_m)| \leq \frac{1}{m}. \tag{79}$$

This allows us to apply another consequence of Mcdiarmid's inequality, which leads to the following, with probability $p \geq 1 - \delta$,

$$\text{err}_P(f) \geq \widehat{\text{err}}_S(f) - \mathop{\mathbb{E}}_{S}\left[ \sup_{f\in\mathcal{F},g\in\mathcal{G}} (\text{err}_S(f) - \widehat{\text{err}}_P(f)) \right] - \sqrt{\frac{\log(1/\delta)}{m}} \tag{80}$$

$$\geq \widehat{\text{err}}_S(f) - \mathcal{R}_m(\mathcal{F},\mathcal{G}) - \sqrt{\frac{\log(1/\delta)}{m}}. \tag{81}$$

This completes the proof.

$\square$

### C.5 PROOF OF THEOREM 5

Now, we outline a set of results pertaining to the joint-entropy like behaviour of $\mathcal{R}_m(\mathcal{F},\mathcal{G})$.

**Theorem 5.** *We are given the mutual complexity measure $\mathcal{I}_m(\mathcal{F},\mathcal{G})$ as defined in co-complexity **P4** (Section 5 of main paper). We consider an alternative ground truth generator space $\mathcal{G}'$, such that $\mathcal{I}_m(\mathcal{G}',\mathcal{G}) = 0$, i.e., $\mathcal{G}$ and $\mathcal{G}'$ are independent. Then we have,*

$$\mathcal{R}_m(\mathcal{F}) \geq \mathcal{I}_m(\mathcal{F},\mathcal{G}) + \mathcal{I}_m(\mathcal{F},\mathcal{G}'). \tag{82}$$

*Proof.* To prove the above, note that it is sufficient to prove the following:

$$\mathcal{R}_m(\mathcal{F},\mathcal{G}) + \mathcal{R}_m(\mathcal{F},\mathcal{G}') \geq \mathcal{R}_m^C(\mathcal{F}) + \mathcal{R}_m^C(\mathcal{G}) + \mathcal{R}_m^C(\mathcal{G}') \tag{83}$$

We elaborate on the left hand side of the above inequality as follows

$$\mathcal{R}_m(\mathcal{F},\mathcal{G}) + \mathcal{R}_m(\mathcal{F},\mathcal{G}') = \frac{1}{2} \times \mathop{\mathbb{E}}_{\sigma,S,S'}\left[ \sup_{f\in\mathcal{F},g\in\mathcal{G}} \left( \frac{1}{m}\sum_{i=1}^{m} f(z_i)f(z_i')g(z_i)g(z_i')\sigma_i \right) \right]$$
$$+ \frac{1}{2} \times \mathop{\mathbb{E}}_{\sigma,S,S'}\left[ \sup_{f\in\mathcal{F},g'\in\mathcal{G}'} \left( \frac{1}{m}\sum_{i=1}^{m} f(z_i)f(z_i')g'(z_i)g'(z_i')\sigma_i \right) \right] \tag{84}$$

We introduce i.i.d Rademacher variables $\{\epsilon_0, ..., \epsilon_m\}$, by multiplying them with the terms within $\mathcal{R}_m(\mathcal{F},\mathcal{G}')$. As $\mathbb{E}[\epsilon_i] = 0 \ \forall i$, this maneuver does not change the value of the expectation.

$$\mathcal{R}_m(\mathcal{F},\mathcal{G}) + \mathcal{R}_m(\mathcal{F},\mathcal{G}') = \frac{1}{2} \times \mathop{\mathbb{E}}_{\sigma,S,S'}\left[ \sup_{f\in\mathcal{F},g\in\mathcal{G}} \left( \frac{1}{m}\sum_{i=1}^{m} f(z_i)f(z_i')g(z_i)g(z_i')\sigma_i \right) \right]$$
$$+ \frac{1}{2} \times \mathop{\mathbb{E}}_{\sigma,\epsilon,S,S'}\left[ \sup_{f\in\mathcal{F},g'\in\mathcal{G}'} \left( \frac{1}{m}\sum_{i=1}^{m} f(z_i)f(z_i')g'(z_i)g'(z_i')\sigma_i\epsilon_i \right) \right] \tag{85}$$

$$\geq \frac{1}{2} \times \mathop{\mathbb{E}}_{\sigma,\epsilon,S,S'}\left[ \sup_{f\in\mathcal{F},g\in\mathcal{G},g'\in\mathcal{G}'} \left( \frac{1}{m}\sum_{i=1}^{m} f(z_i)f(z_i')\left(g(z_i)g(z_i') + \epsilon_i g'(z_i)g'(z_i')\right)\sigma_i v_i \right) \right] \tag{86}$$

Now note that one can re-express $g(z_i)g(z_i') + \epsilon_i g'(z_i)g'(z_i')$ as $(1 + \epsilon_i g(z_i)g(z_i')g'(z_i)g'(z_i'))v_i$. Here $v_i \in \{-1,1\} \ \forall i$. Note that the value of $v_i$ depends on the values of $g(z_i), g(z_i'), g'(z_i), g'(z_i')$ and $\epsilon_i$. However, note that with so many dependencies $v_i$ ultimately is independent of all of these terms, individually. We have,

$$\mathcal{R}_m(\mathcal{F},\mathcal{G}) + \mathcal{R}_m(\mathcal{F},\mathcal{G}') \geq$$
$$\frac{1}{2} \times \mathop{\mathbb{E}}_{\sigma,\epsilon,S,S'}\left[ \sup_{f\in\mathcal{F},g\in\mathcal{G},g'\in\mathcal{G}'} \left( \frac{1}{m}\sum_{i=1}^{m} f(z_i)f(z_i')\sigma_i v_i \right. \right.$$
$$\left. \left. + \frac{1}{m}\sum_{i=1}^{m} f(z_i)f(z_i')g(z_i)g(z_i')g'(z_i)g'(z_i')\epsilon_i\sigma_i v_i \right) \right]. \tag{87}$$

Note that the expectation is over a fixed values of $\sigma, \epsilon, S$ and $S'$. For the next step, we recognize that $v_i \sigma_i = \sigma_i'$ is a Rademacher variable itself in the first term to the R.H.S of the inequality above, as $v_i$ is independent of $f(z_i)$ and $f(z_i')$. For a fixed value of those parameters, we then consider

$$f^* = \arg\max_{f \in \mathcal{F}} \left( \frac{1}{m} \sum_{i=1}^m f(z_i) f(z_i') \sigma_i' \right). \tag{88}$$

Then, we can compute a lower bound for the expression in (87) as follows,

$$\mathcal{R}_m(\mathcal{F}, \mathcal{G}) + \mathcal{R}_m(\mathcal{F}, \mathcal{G}') \geq$$

$$\frac{1}{2} \times \mathop{\mathbb{E}}_{\sigma', S, S'} \frac{1}{m} \sum_{i=1}^m f^*(z_i) f^*(z_i') \sigma_i'$$

$$+ \mathop{\mathbb{E}}_{\sigma, \epsilon, S, S'} \left[ \sup_{f \in \mathcal{F}, g \in \mathcal{G}, g' \in \mathcal{G}'} \left( \frac{1}{m} \sum_{i=1}^m f(z_i) f(z_i') g(z_i) g(z_i') g'(z_i) g'(z_i') \epsilon_i \sigma_i v_i \right) \right], \tag{89}$$

$$\geq \mathcal{R}_m^C(\mathcal{F}) + \frac{1}{2} \mathop{\mathbb{E}}_{\sigma, \epsilon, S, S'} \left[ \sup_{g \in \mathcal{G}, g' \in \mathcal{G}'} \left( \frac{1}{m} \sum_{i=1}^m g(z_i) g(z_i') g'(z_i) g'(z_i') \left( \epsilon_i \sigma_i v_i f^*(z_i) f^*(z_i') \right) \right) \right]. \tag{90}$$

It is important to note here that the variable $\epsilon_i \sigma_i v_i f^*(z_i) f^*(z_i')$ is a Rademacher variable, but however, is not yet independent of the other terms being multiplied to it. The terms $\epsilon_i$, $\sigma_i$, $f^*(z_i)$ and $f^*(z_i')$ are all independent of $g(z_i), g(z_i'), g'(z_i) \& g'(z_i')$, but however, the same cannot be said of $v_i$. Therefore, to resolve this issue we note that the product $g(z_i) g(z_i') g'(z_i) g'(z_i')$ itself can be expressed as another function $g''(z_i, z_i') = g(z_i) g(z_i') g'(z_i) g'(z_i')$, as an instance of another function space $\mathcal{G}''$ and the second term in (89) can then be re-expressed as,

$$\frac{1}{2} \times \mathop{\mathbb{E}}_{\sigma, \epsilon S, S'} \left[ \sup_{g'' \in \mathcal{G}''} \left( \frac{1}{m} \sum_{i=1}^m g''(z_i, z_i') \left( \epsilon_i \sigma_i v_i f^*(z_i) f^*(z_i') \right) \right) \right]. \tag{91}$$

Note that although $v_i$ was not independent of the $g(z_i), g(z_i'), g'(z_i) \& g'(z_i')$, it is independent of the product $g(z_i) g(z_i') g'(z_i) g'(z_i')$, and thus independent of $g''(z_i, z_i')$. Thus, now we can finally express the variables $\{\sigma_i' = (\epsilon_i \sigma_i v_i f^*(z_i) f^*(z_i')) \,|\, i = 1, 2, \cdots, m\}$ as i.i.d Rademacher variables themselves. This leads to the lower bound as follows:

$$\mathcal{R}_m(\mathcal{F}, \mathcal{G}) + \mathcal{R}_m(\mathcal{F}, \mathcal{G}') \geq$$

$$\geq \mathcal{R}_m^C(\mathcal{F}) + \frac{1}{2} \times \mathop{\mathbb{E}}_{\sigma', S, S'} \left[ \sup_{g \in \mathcal{G}, g' \in \mathcal{G}'} \left( \frac{1}{m} \sum_{i=1}^m g(z_i) g(z_i') g'(z_i) g'(z_i') \left( \sigma_i' \right) \right) \right]. \tag{92}$$

$$= \mathcal{R}_m^C(\mathcal{F}) + \mathcal{R}_m(\mathcal{G}, \mathcal{G}') = \mathcal{R}_m^C(\mathcal{F}) + \mathcal{R}_m^C(\mathcal{G}) + \mathcal{R}_m^C(\mathcal{G}'). \tag{93}$$

The last step follows from the fact that $\mathcal{I}_m(\mathcal{G}', \mathcal{G}) = 0$, that is, the function spaces $\mathcal{G}$ and $\mathcal{G}'$ are independent (i.e. $\mathcal{R}_m(\mathcal{G}, \mathcal{G}') = \mathcal{R}_m^C(\mathcal{G}) + \mathcal{R}_m^C(\mathcal{G}')$). This concludes our proof. □

We now prove a corollary to Theorem 5, where we discuss implications of Theorem 5 on the generalization error bound in Theorem 3.

**Corollary 5.1.** *Given a function space $\mathcal{G}'$, such that $\mathcal{I}_m(\mathcal{G}', \mathcal{G}) = 0$. Suppose that $\mathcal{R}_m(\mathcal{F}) = \mathcal{I}_m(\mathcal{F}, \mathcal{G}) + \mathcal{I}_m(\mathcal{F}, \mathcal{G}') + \epsilon$, for some $\epsilon \geq 0$. We have with probability $p \geq 1 - \delta$,*

$$err_P(f) \leq \widehat{err}_S(f) + \mathcal{R}_m(\mathcal{G}) + \mathcal{I}_m(\mathcal{F}, \mathcal{G}') + \epsilon + \sqrt{\frac{\log(1/\delta)}{m}}. \tag{94}$$

*Proof.* This is a direct consequence of the fact that $\mathcal{R}_m(\mathcal{F}) \geq \mathcal{I}_m(\mathcal{F}, \mathcal{G}) + \mathcal{I}_m(\mathcal{F}, \mathcal{G}')$, combined with the generalization bound in Theorem 3. □

Now we proceed to the final set of results, where we decompose the co-complexity into two other complexity terms, as defined in Section 5 of the main paper. First, we show how the co-complexity measure $\mathcal{R}_m(\mathcal{F}, \mathcal{G})$ can be decomposed into $\mathcal{R}_m^I(\mathcal{F}, \mathcal{G})$ and $\mathcal{R}_m^D(\mathcal{F}, \mathcal{G})$.

### C.6  PROOF OF THEOREM 6

The following results discuss the impact of dissociation co-complexity and invariance co-complexity on generalization error bounds.

**Lemma 2.** *Consider function spaces $\mathcal{F}$ and $\mathcal{G}$, such that $\mathcal{F}$ is Rademacher smooth w.r.t. $\mathcal{G}$. Let us define $\mathcal{R}_m^{C,D}(\mathcal{G})$ as $\frac{1}{2} \times \mathbb{E}_{\sigma,S,S',S' \notin \mathcal{I}_{\mathcal{G}}(S)} \left[ \sup_{g \in \mathcal{G}} \left( \frac{1}{m} \sum_{i=1}^m g(z_i)g(z_i')\sigma_i \right) \right]$. For some non-negative real constant $0 \le \alpha \le 1$, we then have*

$$\mathcal{R}_m(\mathcal{F}, \mathcal{G}) \le \alpha \mathcal{R}_m^I(\mathcal{F}, \mathcal{G}) + (1 - \alpha)(\mathcal{R}_m^D(\mathcal{F}, \mathcal{G}) + \mathcal{R}_m^{C,D}(\mathcal{G})), \tag{95}$$

*where $\mathcal{R}_m^I(\mathcal{F}, \mathcal{G})$ and $\mathcal{R}_m^D(\mathcal{F}, \mathcal{G})$ are the invariance co-complexity and the dissociation co-complexity, respectively.*

*Proof.* As the functions are Rademacher smooth, we can decompose the expectation over $S, S'$ to only the ones where $S' \in \mathcal{I}_{\mathcal{G}}(S)$ , and the ones where $S' \notin \mathcal{I}_{\mathcal{G}}(S)$. We begin from (67) as follows.

$$\mathcal{R}_m(\mathcal{F}, \mathcal{G}) = \mathbb{E}_{\sigma,S,S'} \left[ \sup_{f \in \mathcal{F}, g \in \mathcal{G}} \left( \frac{1}{m} \sum_{i=1}^m \frac{(f(z_i)g(z_i)f(z_i')g(z_i'))\,\sigma_i}{2} \right) \right] \tag{96}$$

Then, using the Rademacher smoothness constraint, we have for a certain $0 \le \alpha \le 1$,

$$\mathcal{R}_m(\mathcal{F}, \mathcal{G}) = \alpha \mathbb{E}_{\sigma,S,S',S' \in \mathcal{I}_{\mathcal{G}}(S)} \left[ \sup_{f \in \mathcal{F}, g \in \mathcal{G}} \left( \frac{1}{m} \sum_{i=1}^m \frac{f(z_i)f(z_i')g(z_i)g(z_i')\sigma_i}{2} \right) \right] +$$

$$(1 - \alpha) \mathbb{E}_{\sigma,S,S',S' \notin \mathcal{I}_{\mathcal{G}}(S)} \left[ \sup_{f \in \mathcal{F}, g \in \mathcal{G}} \left( \frac{1}{m} \sum_{i=1}^m \frac{f(z_i)f(z_i')g(z_i)g(z_i')\sigma_i}{2} \right) \right] \tag{97}$$

$$= \alpha \mathbb{E}_{\sigma,S,S',S' \in \mathcal{I}_{\mathcal{G}}(S)} \left[ \sup_{f \in \mathcal{F}} \left( \frac{1}{m} \sum_{i=1}^m \frac{f(z_i)f(z_i')\sigma_i}{2} \right) \right] +$$

$$(1 - \alpha) \mathbb{E}_{\sigma,S,S',S' \notin \mathcal{I}_{\mathcal{G}}(S)} \left[ \sup_{f \in \mathcal{F}, g \in \mathcal{G}} \left( \frac{1}{m} \sum_{i=1}^m \frac{f(z_i)f(z_i')g(z_i)g(z_i')\sigma_i}{2} \right) \right] \tag{98}$$

$$\le \alpha \mathbb{E}_{\sigma,S,S',S' \in \mathcal{I}_{\mathcal{G}}(S)} \left[ \sup_{f \in \mathcal{F}} \left( \frac{1}{m} \sum_{i=1}^m \frac{(1 - f(z_i)f(z_i'))\sigma_i}{2} \right) \right] + (1 - \alpha)(\mathcal{R}_m^D(\mathcal{F}, \mathcal{G}) + \mathcal{R}_m^{C,D}(\mathcal{G})) \tag{99}$$

$$\le \alpha \mathbb{E}_{S,S',S' \in \mathcal{I}_{\mathcal{G}}(S)} \left[ \sup_{f \in \mathcal{F}} \left( \frac{1}{m} \sum_{i=1}^m \frac{(1 - f(z_i)f(z_i'))}{2} \right) \right] + (1 - \alpha)(\mathcal{R}_m^D(\mathcal{F}, \mathcal{G}) + \mathcal{R}_m^{C,D}(\mathcal{G})) \tag{100}$$

$$= \alpha \mathcal{R}_m^I(\mathcal{F}, \mathcal{G}) + (1 - \alpha)(\mathcal{R}_m^D(\mathcal{F}, \mathcal{G}) + \mathcal{R}_m^{C,D}(\mathcal{G})). \tag{101}$$

We use the fact that $\mathbb{E}_{\sigma,S,S',S' \notin \mathcal{I}_{\mathcal{G}}(S)} \left[ \sup_{f \in \mathcal{F}, g \in \mathcal{G}} \left( \frac{1}{m} \sum_{i=1}^m \frac{f(z_i)f(z_i')g(z_i)g(z_i')\sigma_i}{2} \right) \right] \le \mathcal{R}_m^D(\mathcal{F}, \mathcal{G}) + \mathcal{R}_m^{C,D}(\mathcal{G})$, which follows trivially from a derivation similar to that of co-complexity **P3** in Lemma 2 $(\mathcal{R}_m(\mathcal{F}, \mathcal{G}) \le \mathcal{R}_m^C(\mathcal{F}) + \mathcal{R}_m^C(\mathcal{G}))$. This completes the proof.

$\square$

Using this, we proceed to the proof of Theorem 6, where we re-express the generalization error bound in terms of the invariance and dissociation co-complexities, including bounds on the training error. For purposes of simplification, we denote $R_m^I(\mathcal{F}, \mathcal{G}), R_m^D(\mathcal{F}, \mathcal{G})$ as $R_m^D$, and $\mathcal{R}_m^{D,1}(\mathcal{F}, \mathcal{G})$ as $\mathcal{R}_m^{D,1}$.

**Theorem 6.** *Consider function spaces $\mathcal{F}$ and $\mathcal{G}$, such that $\mathcal{F}$ is Rademacher smooth w.r.t. $\mathcal{G}$ and $\mathcal{F}$ contains the constant function $f(z) = c, \forall z, c \in \{-1, 1\}$. Let $err_P$ denote the generalization error for the functions $f \in \mathcal{F}$ which showcase the best fit on the training data samples. For some non-negative real constants $0 \leq \alpha, \beta \leq 1$ and $0 < \delta < 1$, with probability $p \geq 1 - \delta$, we have*

$$err_P \leq (1 - \beta) \left( \frac{1}{2} - \mathcal{R}_m^{D,1} \right) + \alpha \mathcal{R}_m^I + (1 - \alpha)(\mathcal{R}_m^D + \mathcal{R}_m^{C,D}(\mathcal{G})) + \sqrt{\frac{2 \log(1/\delta)}{m}} \quad (102)$$

Note: Recall that $\mathcal{R}_m^{C,D}(\mathcal{G})$ is defined in Lemma 2.

*Proof.* We note that $err_P$ is the generalization error for the case when only the best fitting function $f_{opt} = \arg_{min f \in \mathcal{F}}(\widehat{err}_S(f))$ is chosen for each training data set $S$, and is averaged over all possible ground truth functions $g \in \mathcal{G}$. Note that the labels in $S$ is subject to change, depending on the choice of $g$. Thus for fixed datapoints $z_1, z_2, .., z_m \in S$, (53) changes to

$$err_P \leq \mathop{\mathbb{E}}_{g \in \mathcal{G}} [\widehat{err}_S(f_{opt})] + \mathop{\mathbb{E}}_{g \in \mathcal{G}} (err_P(f_{opt}) - \widehat{err}_S(f_{opt})) \quad (103)$$

$$\leq \mathop{\mathbb{E}}_{g \in \mathcal{G}} [\widehat{err}_S(f_{opt})] + \sup_{g \in \mathcal{G}} (err_P(f_{opt}) - \widehat{err}_S(f_{opt})) \quad (104)$$

$$\leq \mathop{\mathbb{E}}_{g \in \mathcal{G}} [\widehat{err}_S(f_{opt})] + \sup_{f \in \mathcal{F}, g \in \mathcal{G}} (err_P(f) - \widehat{err}_S(f)). \quad (105)$$

We now denote $\phi'(S) = \mathbb{E}_{g \in \mathcal{G}}[\inf_{f \in \mathcal{F}} (\widehat{err}_S(f_{opt}))] + \sup_{f \in \mathcal{F}, g \in \mathcal{G}} (err_P(f) - \widehat{err}_S(f))$. Let us also denote $\psi(S) = \mathbb{E}_{g \in \mathcal{G}}[\inf_{f \in \mathcal{F}} (\widehat{err}_S(f_{opt}))]$ and $\gamma(S) = \sup_{f \in \mathcal{F}, g \in \mathcal{G}} (err_P(f) - \widehat{err}_S(f))$. We then have the following for a fixed $f_{opt}$.

$$\sup_{z_1, ..., z_n, z_i' \in \mathcal{Z}} |\phi'(z_1, ..., z_i, ...., z_m) - \phi'(z_1, ..., z_i', ...., z_m)|$$

$$\leq \sup_{z_1, ..., z_n, z_i' \in \mathcal{Z}} |\psi(z_1, ..., z_i, ...., z_m) - \psi(z_1, ..., z_i', ...., z_m)|$$

$$+ \sup_{z_1, ..., z_n, z_i' \in \mathcal{Z}} |\gamma(z_1, ..., z_i, ...., z_m) - \gamma(z_1, ..., z_i', ...., z_m)| \quad (106)$$

$$\leq \frac{1}{m} + \frac{1}{m} = \frac{2}{m}. \quad (107)$$

It is easy to show that $\sup_{z_1, z_2, ..., z_n, z_i' \in \mathcal{Z}} |\psi(z_1, z_2, ..., z_i, ...., z_m) - \psi(z_1, z_2, ..., z_i', ...., z_m)| \leq \frac{1}{m}$ and $\sup_{z_1, z_2, ..., z_n, z_i' \in \mathcal{Z}} |\gamma(z_1, z_2, ..., z_i, ...., z_m) - \gamma(z_1, z_2, ..., z_i', ...., z_m)| \leq \frac{1}{m}$. Applying Mcdiarmid's inequality, we have with probability $p \geq 1 - \delta$,

$$err_P \leq \mathop{\mathbb{E}}_{S, g \in \mathcal{G}} [\widehat{err}_S(f_{opt})] + \mathop{\mathbb{E}}_S \left[ \sup_{f \in \mathcal{F}, g \in \mathcal{G}} (err_P(f) - \widehat{err}_S(f)) \right] + \sqrt{\frac{2 \log(1/\delta)}{m}}. \quad (108)$$

We already have that $\mathbb{E}_S \left[ \sup_{f \in \mathcal{F}, g \in \mathcal{G}} (err_P(f) - \widehat{err}_S(f)) \right] \leq \mathcal{R}_m(\mathcal{F}, \mathcal{G}) \leq \alpha \mathcal{R}_m^I + (1 - \alpha)(\mathcal{R}_m^D + \mathcal{R}_m^{C,D}(\mathcal{G}))$, for a certain $0 \leq \alpha \leq 1$ (Lemma 2). The first term (expected training error) can be expressed as,

$$\mathop{\mathbb{E}}_{S, g \in \mathcal{G}} [\widehat{err}_S(f_{opt})] = \frac{1}{2} \left( 1 - \mathop{\mathbb{E}}_{S, g \in \mathcal{G}} \left[ \sup_{f \in \mathcal{F}} \left( \frac{1}{m} \sum_i f(z_i) g(z_i) \right) \right] \right). \quad (109)$$

As iterated in the main statement of this theorem, we assume that the classifier function space is complex enough to allow for an error rate $\leq 50\%$ (chance level for binary classification) on the training data itself. This indicates that we can safely assume that $\sup_{f \in \mathcal{F}} \left( \frac{1}{m} \sum_i f(z_i) g(z_i) \right) \geq 0$. We then have,

$$\sup_{f\in\mathcal{F}}\left(\frac{1}{m}\sum_i f(z_i)g(z_i)\right)\sup_{f\in\mathcal{F}}\left(\frac{1}{m}\sum_i f(z_i)g(z_i)\right)$$

$$=\sup_{f\in\mathcal{F}}\left(\frac{1}{m^2}\sum_{i,j}f(z_i)g(z_i)f(z_j)g(z_j)\right) \tag{110}$$

$$=\sup_{f\in\mathcal{F}}\left(\frac{1}{m^2}\sum_{i,j}f(z_i)g(z_i)f(z')g(z')f(z')g(z')f(z_j)g(z_j)\right) \tag{111}$$

$$=\left(\sup_{f\in\mathcal{F}}\left(\frac{1}{m}\sum_i f(z_i)g(z_i)f(z')g(z')\right)\right)^2 \tag{112}$$

where we make use of the fact that $f(z')g(z')f(z')g(z') = 1$. Here, the datapoint $z'$ is sampled from the same distribution $P$. Thus we have,

$$\mathbb{E}_{S,z',g\in\mathcal{G}}\left[\sup_{f\in\mathcal{F}}\left(\frac{1}{m}\sum_i f(z_i)g(z_i)\right)\right] \geq \mathbb{E}_{S,z',g\in\mathcal{G}}\left[\sup_{f\in\mathcal{F}}\left(\frac{1}{m}\sum_i f(z_i)g(z_i)f(z')g(z')\right)\right]. \tag{113}$$

As before, we assume that the function space $f$ is Rademacher smooth, i.e. for a certain $0 \leq \beta \leq 1$, we can express

$$\mathbb{E}_{S,z',g\in\mathcal{G}}\left[\sup_{f\in\mathcal{F}}\left(\frac{1}{m}\sum_i f(z_i)g(z_i)f(z')g(z')\right)\right] =$$

$$=\beta\,\mathbb{E}_{S,z',S\in\mathcal{I}_\mathcal{G}(z'),g\in\mathcal{G}}\left[\sup_{f\in\mathcal{F}}\left(\frac{1}{m}\sum_i f(z_i)f(z')\right)\right]$$

$$+(1-\beta)\,\mathbb{E}_{S,z',S\notin\mathcal{I}_\mathcal{G}(z'),g\in\mathcal{G}}\left[\sup_{f\in\mathcal{F}}\left(\frac{1}{m}\sum_i f(z_i)g(z_i)f(z')g(z')\right)\right]. \tag{114}$$

$$\geq\beta\,\mathbb{E}_{S,z',S\in\mathcal{I}_\mathcal{G}(z'),g\in\mathcal{G}}\left[\sup_{f\in\mathcal{F}}\left(\frac{1}{m}\sum_i f(z_i)f(z')\right)\right]$$

$$+(1-\beta)\,\mathbb{E}_{S,z',S\notin\mathcal{I}_\mathcal{G}(z'),\sigma}\left[\sup_{f\in\mathcal{F}}\left(\frac{1}{m}\sum_i f(z_i)f(z')\sigma_i\right)\right]. \tag{115}$$

The last step above involves the reasonable assumption that the function space is going to be on average worse at fitting random noise labels, rather than a labelling provided by an element of the generator space $\mathcal{G}$. Finally, we note that

$$\mathbb{E}_{S,z',S\in\mathcal{I}_\mathcal{G}(z'),g\in\mathcal{G}}\left[\sup_{f\in\mathcal{F}}\left(\frac{1}{m}\sum_i f(z_i)f(z')\right)\right] = 1, \tag{116}$$

simply by choosing the constant function $f(z) = c$, which is assumed to be present within the classifier's function space $\mathcal{F}$. Also, note that $\mathbb{E}_{S,z',S\notin\mathcal{I}_\mathcal{G}(z'),\sigma}\left[\sup_{f\in\mathcal{F}}\left(\frac{1}{m}\sum_i f(z_i)f(z')\sigma_i\right)\right] = 2\,\mathcal{R}_m^{D,1}(\mathcal{F},\mathcal{G})$. We then have

$$\mathbb{E}_{S,z',g\in\mathcal{G}}\left[\sup_{f\in\mathcal{F}}\left(\frac{1}{m}\sum_i f(z_i)g(z_i)f(z')g(z')\right)\right] \geq \beta + 2(1-\beta)\,\mathcal{R}_m^{D,1}(\mathcal{F},\mathcal{G}). \tag{117}$$

This leads to the final upper bound for $\text{err}_P$ by replacing the terms in (108) by the above. With probability $p \geq 1 - \delta$, we then have,

$$\text{err}_P \leq (1 - \beta) \left( \frac{1}{2} - \mathcal{R}_m^{D,1} \right) + \alpha \mathcal{R}_m^I + (1 - \alpha)(\mathcal{R}_m^D + \mathcal{R}_m^{C,D}(\mathcal{G})) + \sqrt{\frac{2 \log(1/\delta)}{m}} \quad (118)$$

$\square$

Finally, we provide the proof of Proposition 1, which gives interpretable bounds on invariance co-complexity and dissociation co-complexity.

**Proposition 1.** *Define the well known growth function of binary valued functions $\mathcal{F}$ as $A = \Pi_{\mathcal{F}}(m) = \max_z |\{f(z_k) : k = 1, 2, ..., m, f \in \mathcal{F}\}|$. Define the invariance-constrained and dissociation-constrained growth functions of $\mathcal{F}$ w.r.t. $\mathcal{G}$ as:*

$$B = \Pi_{\mathcal{F}|\mathcal{G}}^I(m) = \max_{z, z', z_i' \in \mathcal{I}_{\mathcal{G}}(z_i), \forall i} |\{f(z_k), f(z_k') : k = 1, 2, ..., m, f \in \mathcal{F}\}|$$

$$C = \Pi_{\mathcal{F}|\mathcal{G}}^D(m) = \max_{z, z', z_i' \notin \mathcal{I}_{\mathcal{G}}(z_i), \forall i} |\{f(z_k), f(z_k') : k = 1, 2, ..., m, f \in \mathcal{F}\}|$$

*Define $\mathcal{R}_m^{I'}(\mathcal{F}, \mathcal{G}) = \frac{1}{2} \times \mathbb{E}_{\sigma, S, S', S' \in \mathcal{I}_{\mathcal{G}}(S)} \left[ \sup_{f \in \mathcal{F}} \left( \frac{1}{m} \sum_{i=1}^m f(z_i) f(z_i') \sigma_i \right) \right]$, where $z_i' \in \mathcal{I}_{\mathcal{G}}(z_i), \forall i$. Note that $\mathcal{R}_m^{I'}(\mathcal{F}, \mathcal{G}) \leq \mathcal{R}_m^I(\mathcal{F}, \mathcal{G})$. Given the definitions above, we have $\mathcal{R}_m^I(\mathcal{F}, \mathcal{G}) \leq \sqrt{\frac{\log B}{2m}} \leq \sqrt{\frac{\log A}{m}}$ and $\mathcal{R}_m^D(\mathcal{F}, \mathcal{G}) \leq \sqrt{\frac{\log C}{2m}} \leq \sqrt{\frac{\log A}{m}}$.*

*Proof.* First we show that $\mathcal{R}_m^{I'}(\mathcal{F}, \mathcal{G}) \leq \mathcal{R}_m^I(\mathcal{F}, \mathcal{G})$. This follows from the fact that

$$\mathcal{R}_m^{I'}(\mathcal{F}, \mathcal{G}) = \frac{1}{2} \times \mathbb{E}_{\sigma, S, S', S' \in \mathcal{I}_{\mathcal{G}}(S)} \left[ \sup_{f \in \mathcal{F}} \left( \frac{1}{m} \sum_{i=1}^m f(z_i) f(z_i') \sigma_i \right) \right] \quad (119)$$

$$= \frac{1}{2} \times \mathbb{E}_{\sigma, S, S', S' \in \mathcal{I}_{\mathcal{G}}(S)} \left[ \sup_{f \in \mathcal{F}} \left( \frac{1}{m} \sum_{i=1}^m (1 - f(z_i) f(z_i')) \sigma_i \right) \right] \quad (120)$$

$$\leq \frac{1}{2} \times \mathbb{E}_{S, S', S' \in \mathcal{I}_{\mathcal{G}}(S)} \left[ \sup_{f \in \mathcal{F}} \left( \frac{1}{m} \sum_{i=1}^m (1 - f(z_i) f(z_i')) \right) \right] = \mathcal{R}_m^I(\mathcal{F}, \mathcal{G}) \quad (121)$$

The proof of the main proposition immediately follows from the application of Massart's finite class lemma, which states that for a finite subset $\mathcal{K}$ of $\mathbb{R}^m$, and Rademacher variables $\sigma_1, \sigma_2, ..., \sigma_m$, we have

$$\mathbb{E}_{\sigma_1, \sigma_2, ..., \sigma_m} \left[ \sup_{\vec{y} \in \mathcal{K}} \sum_i \sigma_i y_i \right] \leq r \sqrt{2 \log |\mathcal{K}|} \quad (122)$$

This immediately leads to the well known result for Rademacher complexity $\mathcal{R}_m(\mathcal{F})$, which states

$$\mathcal{R}_m(\mathcal{F}) \leq \sqrt{\frac{2 \log \Pi_{\mathcal{F}}(m)}{m}}. \quad (123)$$

The result above directly follows from the fact that the cardinality of the finite subset $\mathcal{K}$ is upper bounded by the growth function $\Pi_{\mathcal{F}}(m)$, when $\mathcal{K}$ is the set of all possible function outputs from functions $f \in \mathcal{F}$, on any $m$ points. In our case, as we have that

$$\mathcal{R}_m^{I'}(\mathcal{F}, \mathcal{G}) = \frac{1}{2} \times \mathbb{E}_{\sigma, S, S'} \left[ \sup_{f \in \mathcal{F}} \left( \frac{1}{m} \sum_{i=1}^m f(z_i) f(z_i') \sigma_i \right) \right], \quad (124)$$

we can directly apply Massart's lemma, by the using the invariance-constrained growth function $\Pi_{\mathcal{F}|\mathcal{G}}^I(m)$ to obtain the first result as follows. First, we choose the finite subset $\mathcal{K}$ such that it

contains all possible outputs of the function product $f(z_i)f(z_i')$ when $S, S'$ is chosen such that each element in $S'$ lies in the invariance extended subset of each corresponding element in $S$. For this $\mathcal{K}$, the cardinality $|\mathcal{K}|$ is bounded above by the previously defined growth function $\Pi^I_{\mathcal{F}|\mathcal{G}}(m)$. This immediately leads to the first result,

$$\mathcal{R}^{I'}_m(\mathcal{F}, \mathcal{G}) \leq \frac{1}{2}\sqrt{\frac{2\log \Pi^I_{\mathcal{F}|\mathcal{G}}(m)}{m}} \tag{125}$$

$$= \sqrt{\frac{\log \Pi^I_{\mathcal{F}|\mathcal{G}}(m)}{2m}}. \tag{126}$$

We can similarly show the equivalent result for dissociation co-complexity $\mathcal{R}^D_m(\mathcal{F}, \mathcal{G})$, to obtain

$$\mathcal{R}^D_m(\mathcal{F}, \mathcal{G}) \leq \frac{1}{2}\sqrt{\frac{2\log \Pi^D_{\mathcal{F}|\mathcal{G}}(m)}{m}} \tag{127}$$

$$= \sqrt{\frac{\log \Pi^I_{\mathcal{F}|\mathcal{G}}(m)}{2m}}. \tag{128}$$

Finally, we recognize that the growth functions $\Pi^I_{\mathcal{F}|\mathcal{G}}(m)$ and $\Pi^D_{\mathcal{F}|\mathcal{G}}(m)$ are bounded above by the cardinality of all possible function values taken in $S, S'$, which in turn is bounded above by $\Pi_{\mathcal{F}}(m) \times \Pi_{\mathcal{F}}(m)$. This implies $\Pi^I_{\mathcal{F}|\mathcal{G}}(m) \leq \Pi_{\mathcal{F}}(m)^2$ and $\Pi^D_{\mathcal{F}|\mathcal{G}}(m) \leq \Pi_{\mathcal{F}}(m)^2$, from which we then have

$$\mathcal{R}^D_m(\mathcal{F}, \mathcal{G}) \leq \sqrt{\frac{\log \Pi^D_{\mathcal{F}|\mathcal{G}}(m)}{2m}} \leq \sqrt{\frac{\log \Pi_{\mathcal{F}}(m)}{m}} \tag{129}$$

and

$$\mathcal{R}^I_m(\mathcal{F}, \mathcal{G}) \leq \sqrt{\frac{\log \Pi^I_{\mathcal{F}|\mathcal{G}}(m)}{2m}} \leq \sqrt{\frac{\log \Pi_{\mathcal{F}}(m)}{m}}. \tag{130}$$

This concludes the proof. $\qquad\square$

## D  Neural Network Architecture Details

Here we present the architectures that were used in our experiments. As mentioned in section 4, four different architectures were tested. In Table 4, we show the layer-wise details of each architecture. Note that all architectures share a similar parameter count, and also evoke a similar dissociation co-complexity as a result (see Table 2 and 3). Note that the input to all these architectures is an image of size $28 \times 28$ (same size as in the MNIST dataset). For STL-10, we resized the input ($96 \times 96$) to $28 \times 28$ and used the same architectures for consistency.

| **2-Layer NN** (15.8k) | **Standard CNN** (16.1k) | **Scale Equivariant CNN** (15.9k) | **Rotation Equivariant CNN** (16k) |
|---|---|---|---|
| FC-20 | Conv-15 (3x3) | ScaleConv-10 (11x11, 2.2:1) | P4ConvZ2-10 (3x3) |
| FC-2 (Output) | Maxpool(3x3) | Maxpool(3x3) | Maxpool (3x3) |
| | Conv-30 (3x3) | ScaleConv-22 (11x11, 2.2:1) | P4ConvP4-15 (3x3) |
| | Maxpool (3x3) | Maxpool(3x3) | Maxpool (3x3) |
| | Conv-40 (3x3) | ScaleConv-32 (11x11, 2.2:1) | P4ConvP4-19 (3x3) |
| | Maxpool (3x3) | Maxpool(3x3) | Maxpool (3x3x4) |
| | FC-20 | Scale-Pool (4x1) | FC-10 |
| | FC-2 (Output) | FC-20 | FC-2(Output) |
| | | FC-2 (Output) | |

Table 4: Shown are the architectural details of all neural networks used in our experiments. Scale-Conv represents scale-equivariant convolution and P4Conv-Z2 and P4Conv-Z4 represents rotation equivariant convolution (as detailed in Cohen & Welling (2016a)). For the ScaleConv layers, note that 2.2:1 represents the ratio of the maximum to the minimum scale of the filters. A total of 4 scale pathways were chosen for those layers. FC represents the fully connected layers. Networks are chosen such that overall they share a similar parametric count (shown within the brackets).

