# OpenReview forum: "Co-complexity: An Extended Perspective on Generalization Error"
_ICLR.cc/2021/Conference — Reject_

### Official Review · AnonReviewer3 · 2020-10-20
**A New Perspective on Generalisation Error**

**Rating:** 4
**Confidence:** 4

**Review:**

** Description

This paper attempts to derive a new set of generalisation bounds using the idea of matching the function space of the learning machine with the function space describing the generation of labels.

** Pros

This may be a mechanism for capturing the fact that generalisation performance will depend on the structure of the data and not just on the complexity of the function space of the learning machine.  This is potentially an important contribution.

** Cons

I find the set up of this framework very confusing.  In the motivation (paragraph 3 of section 2) we are invited to imagine a random function with a very good fit to the training data and a low complexity measure.  Complexity measures tell us that with overwhelming probability a low complexity will not fit the training data well provided we have a sufficiently large training set.  What we would expect is that a low complexity machine would do little better than chance so the generalisation gap is small.  It is difficult to give this paper much credence after paragraph.

In the discussion around Figure 1 there seems to be a confusion between generalisation performance and generalisation gap.  A linear separable function will with high probability have a much lower generalisation error for a perceptron than learning a completely unstructured function, but the complexity provides guarantees on the generalisation gap not on the generalisation performance.

I am deeply puzzled by your comments that traditional bounds are optimistic (e.g. Remark 1).  What is the learning scenario when the label generating function is not fixed.  Clearly the function being learned is not known. it may belong to a large class of functions, with known constraints but in normal classification it is fixed.  I cannot make sense of the learning framework you are modelling.

I don't know what your experimental section is trying to say.  It sits awkwardly with the theorems and doesn't, in my view, really says anything useful.  Or, at least, this is not well explained in the text.

** And Yet...

I think there is potentially a lot to be gained from this approach. The generalisation performance does depend on the problem being learned and not just on the architecture of the learning machine (or complexity of function space F).  This work points to a way to incorporate this information.  The joint Rademacher complexity might provide a useful way of expressing how attuned a learning machine is to the problem being learned.  I just think the story around the theorems is confused.  In my judgement this work is probably too immature for ICLR.  Maybe I am being dull and with minor modifications the authors can make this paper more coherent and allay my doubts.  With a clearer explanation of what these theorems are telling us I believe this could
be an important contribution to the field, but in it present form I think the paper adds more confusion than light.

---

> ### Author Response · Authors · 2020-11-20
> **Response to Reviewer 3 (Part 1)**
>
> We appreciate your comments and observations on the motivation of our approach. The following are our responses and clarifications.  First, we would like to mention that the motivation section is being thoroughly clarified and changed in accordance with your comments. The final version with the appropriate changes will be uploaded soon.
>
> 1.   “Complexity measures tell us that with overwhelming probability .... generalisation gap is small.”
>
> R: In this paper we are making the argument that the complexity of the generator space has a role in quantifying the true generalization gap of a classifier (i.e, depends on the problem being learned). In the motivation, we try to make this point by considering random label generating functions (LGF) and low-complexity classifiers.  In this case, while it is true that a low complexity classifier may not fit a typical instance of training data well, there are (with non-zero probability) instances of training data where there exist low-complexity classifiers that show a significantly better fit than chance. For all these instances, the Rademacher bound would still predict a small generalization gap, and thus a small generalization error (as training fit in all those instances is good). However, the true generalization gap in those cases will be very large as the LGF is unstructured. This problem can be addressed by adding the complexity measure of the underlying generator space (and also subsequently quantifying the similarities between the classifier and generator spaces).
>
> In the paper (section B.3 of the Appendices) we demonstrate an example with 2-layer neural network LGFs and 1-layer neural network classifiers, to showcase the above aspect. There, we find that when the generator space gets incrementally more complex/larger (neural networks with more hidden neurons), the generalization gap also is greater in direct proportion to the complexity of the generator space. Note that more complex generator spaces negatively affect training performance, as you observed, but because there are more degrees of freedom on the LGF’s behavior on the test data, the observed generalization gap is also larger. Thus here, as the generator neural network grows in complexity with more hidden neurons, while both the training and test performance of the classifier (a single-layer neural network) suffer negatively, the difference in test and training performance grows in direct proportion to the complexity of the generator. The original Rademacher bound cannot capture this trend as it stays fixed for all cases.
>
> 2.   “In the discussion around Figure 1 ...... not on the generalisation performance.”
>
> R: Our new bounds (Theorems 2 & 3) also provide guarantees on the generalisation gap. Fig. 1 shows two cases with the same classifier function space but different generator spaces. In each case, the training data instance is such that there exists a classifier that has good fit on the training data (which will happen with non-zero probability), but different generalization performance due to the difference in the LGFs. By doing so, the generalization performance in these two cases is solely dictated by the generalization gap.

---

> > ### Author Response · Authors · 2020-11-20
> > **Response to Reviewer 3 (Part 2)**
> >
> > 3.  “I am deeply puzzled by your comments ...... learning framework you are modelling.”
> >
> > R:  The learning framework we are using is no different from the standard learning framework. The main point of our work is that problem specific constraints play a role in understanding the generalisation gap. The aim of learning is to approximate the true label generating function (which we agree is fixed but unknown). What we are given is problem specific constraints (such as translation invariance in vision problems), as well as the value of the LGF on a set of training data. Subject to these constraints, the LGF belongs to some space we call the generator space ($G$), i.e. it can be any of the functions within $G$. Note that the generator space arises as a consequence of the unknowability of the LGF. It is well known that the complexity of the classifier function space ($F$)  is involved in bounding the generalization gap. We argue that the complexity of the generator space also plays a role in bounding the generalization gap. We show that the upper bound on the generalization gap now should be dependent on the supremum over all functions within $F$ which maximize the generalization gap, and also on the supremum over all functions within $G$ which maximize the generalization gap. Therefore, with this interpretation, we find that the generalization gap depends on the complexity of both the classifier and the generator function spaces.
> >
> > Note that generator spaces can also be used as a tool to represent the level of structuredness expected in the LGF: if we know that the LGF is highly unstructured due to the lack of any pre-known constraints, it potentially belongs to a larger set of functions and thus results in a more complex generator space (thus, harder to generalize for classifiers). On the other hand if it is known that the LGF is very structured and therefore heavily constrained (e.g. linear), then it belongs to a much smaller set of functions and thus evokes a less complex generator space (easier to generalize).
> >
> > 4.    “I don't know what your experimental ..... in the text.”
> >
> > R:  The experiments in the main paper are all aimed at testing the implications of Theorem 6, which we believed to be our most significant contribution: co-complexity can be broken into two separate terms, and reducing the invariance co-complexity while maintaining dissociation co-complexity is guaranteed to improve generalization performance (maintain training performance and improve test performance). To test this, we evaluate state-of-the-art network architectures which have different levels of invariance co-complexity (NN, CNN, Scale-Equivariant CNN, Rotation-Equivariant CNN) to common visual transformations. Our main finding from the experiments is that while architecture choices (max-pooling/scale-invariance/rotation-invariance) present in CNNs and their variants reduce their invariance co-complexity w.r.t the classification problem, the dissociation co-complexities of these networks are unaffected. Thus, while these networks are more adept at being invariant to common visual transformations, they do not lose the ability to discriminate between two separate visual categories. This result is relevant to Remark 7, which says that ideally we should construct classifiers which have lower invariance co-complexity ($R_m^I$) while maintaining their dissociation co-complexity ($R_m^D$), and this cannot be simply achieved by reducing the Rademacher complexity of the classifier.
> >
> >
> > 5.    “With a clearer explanation of what these theorems ...... confusion than light.”
> >
> > R:  We hope we have convinced you that the generalization performance depends on both the classifier function space (the so called learning machine) and the generator function space (essentially the problem being learned).

---

### Official Review · AnonReviewer1 · 2020-10-26
**Official Blind Review #1**

**Rating:** 5
**Confidence:** 4

**Review:**

This paper studied a novel perspective on generalization error bounds, by introducing the "label generating function "(LGF). Several new complexity measures (correlated Rademacher complexity, co-complexity, invariance co-complexity, dissociation co-complexity, Rademacher smoothness) were proposed. The properties of the measures and generalization error bound with respect to these complexity measures are studied.

This paper is written in clear form and easy to follow.  There are some points I did not fully understand.

1. In formula (4) , the invariance classed of G contains n functions.  However, there may be uncountably infinitely many function \tau's satisfying g(\tau(z)) = g(z). In this case, how is I_G(z) in (5) defined? And the definition R_m^I(F, G) and R_m^D(F, G) rely on z'_i in I_G(z_i) or z'_i not in I_G(z_i), but what is the distribution of z'_i over I_G(z_i) (for finite I_G(z_i), countably infinite I_G(z_i) and uncountably infinite I_G(z_i) respectively)? I think I did not fully understand this part so I could not understand Theorem 5 and Theorem 6 in Section 6.

2. Theorem 1 of Section 6 holds true for \hat{err}_S(f) = \sum_i (f(z_i)/m) and err_P(f) = E[f(z)]. Does it also hold true if we introduce g in the definitions of \hat{err}_S(f) and err_P(f) in formulas (9) and (10)? If it still holds true, then would Theorem 2 just mean nothing because it is just a looser upper bound of Theorem 1?

3. If we consider h=1-f*g and consider the Radamacher complexity of function h in Theorem 1, we can have an upper bound which is a simple corollary of Theorem 1. What is the relationship of this corollary and Theorem 2-4?

With these unclear questions, I vote for a marginal reject. I would like to change my score if these questions are well addressed by the authors.

---

> ### Author Response · Authors · 2020-11-20
> **Response to Reviewer 1**
>
> Thanks for your comments and questions. Here are our responses and clarifications to the same. Changes are being made to the main paper to clarify these points as well.
> 1.   The current definition indeed only pertains to the case where the cardinality of the invariance extended set $I_G(z_i)$  of a point $z_i$ is countably infinite. It turns out that we can redefine relevant quantities so that the formulation applies in the uncountably infinite case. Details are below.
> To resolve this, we now define $I_G(z_i)$ as the set of all $z$ in $\mathbb{R}^d$, such that $z$ relates to $z_i$ via some invariance transformation. Note that this set can be uncountably infinite in nature as well. The set can also be now represented via a simple indicator function $I(z,z_i)$, which is 1 when $z$ is related to $z_i$ via some invariance transformation, and 0 otherwise. Subsequently, in the definitions of invariance and dissociation co-complexities, $z’_i$ is sampled according to the distribution $P(z)I(z,z_i)$ (note that this will add to less than 1, thus will need normalization) and $P(z)(1-I(z,z_i))$ respectively, where P is the original data distribution (also defined in the main paper, section 5 para 1). Thus, the distribution of $z’_i$ is conditioned only on $z_i$, and is non-zero only in the invariance extended set of $z_i$ for invariance co-complexity, and the opposite for dissociation co-complexity.
>
> 2.   Theorem 1 of our main paper does hold true when $g$ is also introduced, but that is because in the conventional setting, $g$ is assumed to be completely known given the training data (the generator space only contains a single element). However, as in this case we assume that the ground truth label generating function (LGF) belongs to a function space $G$ created from known problem-specific constraints, we find that additional terms need to be introduced. To show this, we primarily exploit the fact that the LGF is always unknown. In this scenario, given an instance of training data samples and their labels,  there still exists an entire set of possible LGFs (denoted by $G_{small}$) within $G$, all of which have the same labels on the given training data. With this observation, we immediately see that the true upper bound of the generalization gap requires us to take the supremum of the generalization gap over all possible $f$ within $F$ and all possible $g$ within $G_{small}$, which can be further bounded by supremum over $G$, as $G_{small}\subseteq G$. Theorem 2 follows from this observation, where we find that the generalization gap depends on the size/complexity of both the classifier’s and generator’s function spaces, in the case when the LGF belongs to one of an entire set of functions within the generator space. Hence, although the additional term of $R_m(G)$ seems to loosen Theorem 1, within the context of problem-specific constraints, we find that the complexity of the generator space also plays a part in quantifying the generalization gap.
> 3.   The corollary which results from $h=1-fg$ would have the same form as Theorem 1, as the function $g$ is usually assumed to be fixed and unchanged. However, if we assume that $g$ belongs to another function space $G$, then Theorem 2 can be derived from Theorem 1. Theorems 3 and 4 however do not follow immediately from Theorem 1 as they require the pairing of corresponding terms $z_i$ and $z’_i$ in $S$ and $S’$, (the proof branches out from the proof of Theorem 1 after the pairing). The reason we choose to pair corresponding terms is to obtain the multiplicative form $f(z)g(z)f(z’)g(z’)$, which simplifies when $z’$ belongs to the invariance extended set of $z$ ($g(z)g(z’)=1$), enabling the decomposition of co-complexity into two different measures. We find this decomposition necessary because in most cases, the generator space can only be represented via knowledge of pre-known constraints, such as invariances to various transformations.

---

### Official Review · AnonReviewer2 · 2020-10-28
**Nontitle's**

**Rating:** 7
**Confidence:** 4

**Review:**

##########################################################################
Summary:

The paper provides an interesting perspective to view generalization error for the machine learning model. In particular, it proposes to investigate the constraint on the label generating function space. They propose a concept of co-complexity analogous to the entropy-ish concept which measures complexities between two function spaces. This co-complexity can be decomposed into two parts which measure the categorization ability of the classifier in generator and extent level in classifier for the invariance transformation in the generator.
##########################################################################
Reasons for score:

Overall, I vote for accepting. I think a reconsideration of the established theory is good. My concern is about the clarity of the paper and experiments (see cons below). Hopefully, the authors can address my concern in the rebuttal period.

##########################################################################Pros:

1. The paper takes an important issue of generalization error estimation. For me, the problem itself is of significance and interest.

2. The proposed method to measure complexity is novel for capturing the relationship between generator and classifier space and how the given constraints affect complexity.

3. This paper provides comprehensive experiments, including both qualitative analysis and quantitative results, to show the effectiveness of the proposed framework.

##########################################################################
Cons:

Although the proposed method provides several ablation studies, I still suggest the authors conduct the following ablation studies to enhance the quality of the paper:
1. The experiment can be better designed to verify the theorems. Probably the proportion of data generated by invariant transforms can be set with different levels so that it can be verified that the second term in co-complexity reflects the change of R_d.
2. A couple of comparisons on certain simple models between extended Rademacher complexity and the original R-complexity can be given to show the better tightness of the new bound.
3. Though this paper is full of theorems, I would be happier to see more insights that can be used in practice. I would appreciate that if the authors can bridge this extended Rademacher complexity to the real guidance on how to train the model with better generalization. E.g. is data augmentation useful to reduce R_I?
4. The paper seems very like a journal paper w.r.t. the length and structure. I am just thinking that the paper's content could be better presented and a better fit with its full journal version.

#########################################################################
Some typos:
(1) abstract: large irrespective -> largely irrespective
(2) Does Eq (6) miss \sigma.
(3) Remark 4 is not coherent to Theorem 5 very well. It needs more clarification.
(4) It is not clear that why R_m^I' m(F; G) <= R_m^I m(F; G)

---

> ### Author Response · Authors · 2020-11-20
> **Response to Reviewer 2**
>
> Thank you for your feedback. We appreciate the insightful comments and suggestions. Here are our initial responses, before we upload the version of the paper with the necessary and suggested changes.
>
> 1. “The experiment can be better... change of $R_m^d$.”
>
> R: This is a very useful suggestion, and we can use it to prove the correspondence between the generalization gap and changes in $R_m^I$ and $R_m^d$. However, currently, we do not yet have an exact analytic expression for the $\alpha$ term in Theorem 6. Thus, in the main paper, we chose to do a comparative study with different networks which have similar values of $R_m^d$, but different values of $R_m^I$, keeping the data and their invariance transformations fixed (fixed $\alpha$). We are working on obtaining estimates/bounds for $\alpha$ which would allow precise computation of the bounds. Currently, the main implication of theorem 6 lies in the fact that constructing classifiers with lower $R_m^I$ while maintaining $R_m^D$ will guarantee better generalization performance.
>
> 2. “A couple of comparisons .... new bound.”
>
> R: We are attempting an experiment where we showcase how the correlated Rademacher complexity can yield tighter bounds for neural networks, compared to Rademacher complexity. However, currently we do not have closed form analytic expressions for correlated Rademacher complexity for neural networks with single hidden layers, and thus we are computing them empirically.
>
>  3. “Though this paper is full of theorems....reduce $R_m^I$?”
>
> R: We are conducting an experiment to obtain deeper insights into how various levels of data augmentation change $R_m^d$ and $R_m^I$ for CNNs and MLPs. Currently, we find that as data augmentation levels increase, expectedly $R_m^I$ almost always reduces, but $R_m^d$ may decline as well if too much augmentation is performed (thus ability to discriminate may suffer). Also, we are adding more explanation and consequences of our theorems with a more detailed discussion. For instance, we note that monitoring trends of invariance and dissociation co-complexity can be useful for architecture selection (especially CNN and its variants).
>
> 4. “The paper seems very like a .... journal version.”
>
> R: We are working towards extensions of our approach with more theoretical and empirical results for a journal version and will try to ensure that the conference and journal version complement each other.
>
> Other comments and typos:
> (2) Eq. 6 does not have $\sigma$.
> (3) We agree, and will be providing a more coherent explanation of Theorem 5's implications.
> (4) We are the adding the proof of this result in the appendices.

---

### Official Review · AnonReviewer4 · 2020-10-29
**Interesting work but, in this form, the paper not ready for publication.**

**Rating:** 4
**Confidence:** 3

**Review:**

This paper aims to propose a new complexity measure, called co-complexity, to control classifiers' generalization gap. This new measure acts like a joint-entropy and leads to tighter bounds on the generalization error in this setting. The main idea is to extend the classical complexity measure of Barlett & Mendelson (2003) by introducing a new function space: the generator space is defined as the function space of all possible LGFs satisfying ad hoc constraints. Thus, the authors claimed to be able to measure the extent to which the classifier's function space obeys the invariance transformations in the data and measure the extent to which the classifier can differentiate between separate categories in the data.

Section 2 aims at justifying the introduction of this generator space, Section 5 gives useful definitions, and Section 6 provides theoretical results. Evidence of these results is reported in the supplementary material. Section 7 presents some experiments, but most of them are in the appendix.

–

As it stands, this work does not seem to me to be ready for publication. Indeed, despite some interesting developments, the presentation needs to be deeply improved. My main reproach concerns Sections 5 and 6: definitions and evidence are chained together without any link between them, little contextualization, explanation, thought process. The experimental part suffers from a great lack of detail. Finally, the writing of the evidence in the appendix can be improved. However, unless I am mistaken, they are accurate (and easy to follow).

Here is a list of suggestions for the authors:
1. The numbering of some paragraphs is weird. In particular, I am quite surprised that Sections 2, 3 and 4 are not subsections of the introduction (Section 1).
2. LGF is not properly defined, whereas it is a central notion.
3. Section 2 is based on the complexity measure $R_m(f)$. However, this concept is only defined in Section 5, two pages later.
4. Section 5 looks like a chain of definitions without motivation and intuition. It is necessary to better motivate the definitions, for example, by providing intuitions about the nature of the objects concerned, especially since there are many of them.
5. Revise the definitions of "co-complexity of invariance" and "co-complexity of dissociation". Currently, it is not clear at first reading whether $z_i'$ are constrained to belong (or not) to $I_G(z_i)$ and that this is not always the case a priori.
6. The definition of "Rademacher smoothness" mixes intuition with mathematical definition, making it difficult to understand.
7. In the definition of Dissociation Co-Complexity, "We also define a variant [...], in which $S$ contains only one instance, instead of $m$ instances, denoted as $R_m^{D,1}(\mathcal{F},\mathcal{G})$" What is the sum then about?
8. As with Section 5, the presentation of Section 6 and the appendices could be improved.
9. Computations 9 and 10 would benefit from being stated with "real" fractions, i.e. displaystyle fraction.
10. As noted above, Appendix A suffers from a lack of context. Besides, there is no mention of this appendix in the text's body, other than in the outline.
11. In Appendix C, several references are made to the definitions in Section 3. I think the authors meant to refer to Section 5.
12. The demonstrations should be shorted, particularly those in Appendix C.1. Evidence (i) is clearly too trivial to be detailed, for example.
13. In many places, the writing could be improved: often, variables are used before being defined. For example, this is the case in Calculation (34), where the $\sigma_i'$ are defined only two lines later. The same is true at several locations in the rest of the appendix.
14. Evidence (iv) page 13: I think there is a parenthesis error when introducing $v_i$. A $\frac12$  is missing from calculations (36) and (37).
15. Many superfluous parentheses in the demonstration of Theorem 2 do the reading of the latter tedious. There are superfluous parentheses in the numerator of fractions in almost all computations: (58), (59), (60), (61), (62), (63), once in the body of the text just after (64), (69). Equation (59), fractions should be presented in extended form. Twice the authors write $[-1,1]$ for $\lbrace-1,1\rbrace$.
16. For me, the main subtlety of the demonstration concerns the multiplication of each of the terms of the sum by $\sigma_i$ (Equation (61)). I am not convinced by the way the authors justify this point. The same argument is used several times thereafter.

Typos:
* Page 4, penultimate line: "that" is doubled without reason;
* Page 5, 2nd paragraph of Section 6: is/be;
* Page 17, just after (72): There is an extra parenthesis in this sentence. On the other hand, the way of introducing the $\sigma_i'$ as a set seems to me unnecessarily cumbersome.

---

> ### Author Response · Authors · 2020-11-20
> **Response to Reviewer 4**
>
> Main Comments:
> We appreciate the detailed feedback on our work, and are making all necessary changes to improve presentation, clarity and motivation within the paper (especially sections 5 and 6). Here are our responses to the queries.
>
> “My main reproach concerns Sections 5 and 6: definitions and evidence are chained together without any link between them, little contextualization, explanation, thought process.”
>
> R: The main idea in this paper is to argue that the generalisation performance will depend on the structure of the data (the ground truth label generating function) and not just on the complexity of the function space of the learning machine. The definitions in Sec 5 are motivated by this idea and build on existing work, namely Rademacher Complexity. In the interest of space, we hope readers have some familiarity with existing work and cite appropriate references from the literature. For the technical results in Sec 6, we have provided explanatory remarks (7 remarks in total), We hope these are sufficient to convey our ideas.
>
>
> “The experimental part suffers from a great lack of detail”.
>
> R: The experiments in the main paper are all aimed at testing the implications of Theorem 6, which we believed to be our most significant contribution: co-complexity can be broken into two separate terms, and reducing the invariance co-complexity while maintaining dissociation co-complexity is guaranteed to improve generalization performance (maintain training performance and improve test performance).
>
> List of Suggestions:
> Below we address several of the main suggestions that require clarifications. Regarding the remaining suggestions and typos, we will make the relevant modifications in the revised paper.
>
> Suggestion 5) We are revising and clarifying the definitions. For invariance co-complexity, $z’_i$ are indeed constrained to be within $I_G(z_i)$, whereas for dissociation co-complexity $z’_i$ is constrained to not be within $I_G(z_i)$ (thus lies in $\mathbb{R}^d-I_G(z_i)$).
>
> Suggestion 6) We are changing the definition of Rademacher smoothness, to a much more concrete and exact formulation.
>
> Suggestion 8) We are adding more context to the implications of the theoretical results, and improving the appendices.
>
> Suggestion 10) We are providing more context and background to the appendix, and more clear references to the same in the main paper as well.
>
> Suggestion 16) Currently we adapt this multiplication step from the initial work on Rademacher complexity, as they observe that multiplying with rademacher random variables $\sigma_i$ does not change the expectation. Here, $\sigma_i$ can be construed to be an indicator of whether $z_i$ and $z’_i$ are put in $S$ and $S’$ or whether they are put in $S’$ and $S$ respectively. Thus we see that multiplication with $\sigma_i$ which follow the Rademacher distribution will keep the expectation unchanged.

---

### Author Response · Authors · 2020-11-25
**Summary of changes made in the revised paper**

We would like to thank all reviewers for their critical reviews and valuable suggestions. Here we summarize the changes made in the revised paper.

**Motivation**: Changed the write-up of section 2, now focusing on how the unknowability of the label generating function can affect the true generalization gap. Improved the presentation of the cases where random label generating functions with and without structure are compared. Adjusted the explanation of how more complex label generating functions will likely lead to a worse generalization gap, in accordance with reviewer 3’s comments on how LGF complexity may affect training error.

**Definitions**: Added more explanation of each proposed complexity metric: correlated Rademacher complexity, Co-Complexity, Invariance and Dissociation Complexity. Changed the definition of Invariance co-complexity in accordance with reviewer 4 and 1’s comments, now allowing for the general case of uncountably infinite invariance extended sets. Also, clarified the sampling of z’_i with more precise formulations for both invariance and dissociation co-complexity. We note that these changes in definition do not affect any of the theoretical results/proofs. Also, clarified the definition of Rademacher smoothness.
Theoretical Remarks: Added more explanation in the remarks of some of the theoretical results, mainly for Theorem 2 and Theorem 5 (reviewer 2’s concern).

**More experimental details**: Added more detail to the MNIST/STL-10 experiments, describing the conversion of the multi-label classification framework into a binary classification problem (necessary for our complexity metrics). Also added more detailed explanation of the various compared architectures, including the exact architectures used in the experiments (Appendix D). Added a figure (figure 3) that shows precise variation in the generalization gap trend in response to change in the number of hidden neurons in the label generating function (2-layer neural network). Results show how the generalization gap precisely depends on the complexity of the generator.

**Appendices**: Implemented all requested modifications to the appendices (reviewer 4), including shortening the proofs wherever possible, fixing technical details, and adding appropriate explanations and assumptions wherever necessary. Now we’re referring to appendix A in the context of our main theorems.


**Other changes that were suggested**: An experiment was suggested by reviewer 2 to demonstrate the tightness of the correlated rademacher complexity bound. We found that without a closed form expression for the correlated rademacher complexity, we are not in a position yet to fairly compare with well known expressions for rademacher complexity. We’re working to find analytic expressions for $R_m^C(F)$ so that in future we can analyze the same.
We were also working on the experiments where we show how data augmentation affects invariance and dissociation co-complexity (as suggested by reviewer 2). Although we have preliminary results, the computed co-complexity metrics are “local” in nature, rather than their proposed “global” definitions. That is because we could only compute these quantities for the subset of classifier functions that are computed after training. We believe that adding a new set of local definitions for all our metrics was beyond the scope of the current paper, and requires more elaborate theoretical treatment.

**Final remarks**: Please note that due to major space constraints (even after extra page), we had to prioritise the suggested modifications that were more critical compared to others. As reviewer 2 had noted, we are indeed working on a journal version of this work which will include the extra experiments and the local analogues of the complexity metrics. We will ensure that the journal and this draft complement each other.

---

### Decision · Program_Chairs · 2021-01-07
**Final Decision**

**Decision:**

Reject

**Comment:**

The paper considers generalization in setups in which the training sample
may be generated by a different distribution than the one genertaing the test data.
This sounds much like transfer learning, and similarly sounding considerations,
of a space of possible generating distributions, ways of measuring the statictical complexity
of such spaces and implied error generalization results were analyzed in e.g.,
Jonathan Baxter's "Theoretical models of learning to learn" 1998 and
S Ben-David, R Schuller "Exploiting task relatedness for multiple task learning"
S Ben-David, RS Borbely "A notion of task relatedness yielding provable multiple-task learning guarantees"
Machine learning 73 (3), 273-287

The current submission does not mention these earlier works.

Furthermore, the paper suffers from mathematical sloppiness. The model uder which the generalization theorems
hold is not clearly defined. For example,  Theorem 2, Theorem 3 and Theorem 4  do not stae what are the probability spaces to which the "probaility p > 1-\delta" quantifications refer.